# NeMoS: Nearest Neighbors Bandit meets Active Learning for Online Model Selection

## Abstract

The proliferation of open-platform text-to-image generative models has made prompt-wise model selection critical to maximize generation quality and semantic alignment. However, current strategies, such as contextual bandits, often converge slowly and fail to exploit the semantic relationships across prompts. To bridge this gap, we propose NeMoS, a non-parametric bandit framework that couples nearest neighbor reward estimation with a budget-constrained active learning strategy. Specifically, our approach operates in the prompt embedding space and estimates the reward of incoming prompts based on feedback from their nearest neighbors. By limiting ground-truth queries to ambiguous "near-tie" scenarios, NeMoS resolves uncertainty efficiently and accelerates convergence. We prove that this active mechanism yields a poly-logarithmic regret bound, marking a significant theoretical improvement over its passive version. Extensive experiments on four datasets with six image generative models show that NeMoS reduces regret by up to 60% compared to state-of-the-art baselines, while being robust to model addition or removal. *We provide experimental code in the supplementary material.*

## 1 Introduction

In recent years, there has been a proliferation of prompt-guided image generation models, with improving fidelity and diversity performance (Ho et al., 2020; Rombach et al., 2022; Reed et al., 2016; Xu et al., 2018; Podell et al., 2024; Ding et al., 2021). As a result, practitioners now face a wide range of available models, each with their own strengths and weaknesses: some models prioritize photorealism, others focus on creativity or speed of inference (Jiang et al., 2025). The principle of one model outperforming all other competitors on any given task thus becomes impossible to achieve in practice. Therefore, the assignment of a given prompt to the best generative model available is a problem both critical and non-trivial.

A conventional approach to this challenge relies on aggregate performance scores, typically computed as averages over a large set of prompts using metrics such as CLIPScore (Hessel et al., 2021) or PickScore (Kirstain et al., 2023). However, these global averaged scores do not reflect potential variations in model performance across different types of prompts. As demonstrated in the illustrative example of Figure 1, the SDXL-Turbo (Podell et al., 2024) text-to-image model achieves the highest CLIPScore on the first prompt, while the model Sana (Xie et al., 2024) offers a higher CLIPScore for the second prompt. These discrepancies in prompt-level model ranking are not exceptions, but rather a common occurrence in generative model behavior. This is the result of different data distributions being used for the training of each model (and often distinct architectures and objectives being employed (Frick et al., 2025)). This observation highlights a crucial shortcoming of global ranking methods and motivates the need for prompt-aware model selection strategies. Furthermore, a static strategy that always defaults to the largest model is computationally wasteful, as it fails to leverage lighter architectures when they suffice.

Model selection has hence recently emerged as a key challenge in generative AI, with offline methods proposing to rank prompts or datasets against candidate models (Luo et al., 2024; Lewandowski et al., 2025). In contrast to these static approaches, PAK-UCB (Hu et al., 2025b) addresses prompt-aware selection in an online setting by formulating it as a contextual bandit problem. In this scenario the learner observes

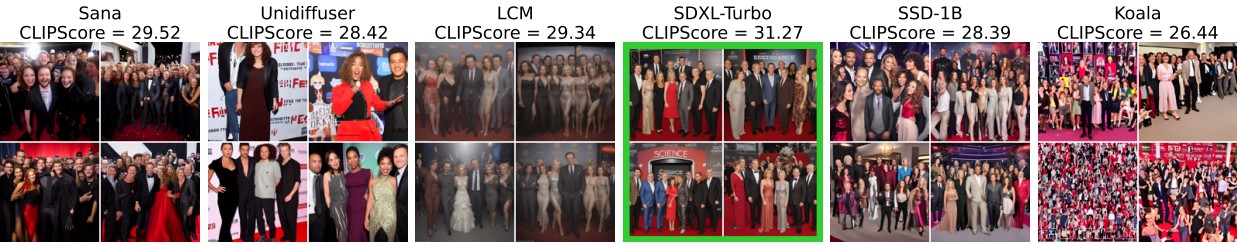

(a) Prompt: "*celebrating their show : actor joins the cast and crew of science fiction tv program for the red carpet event*"

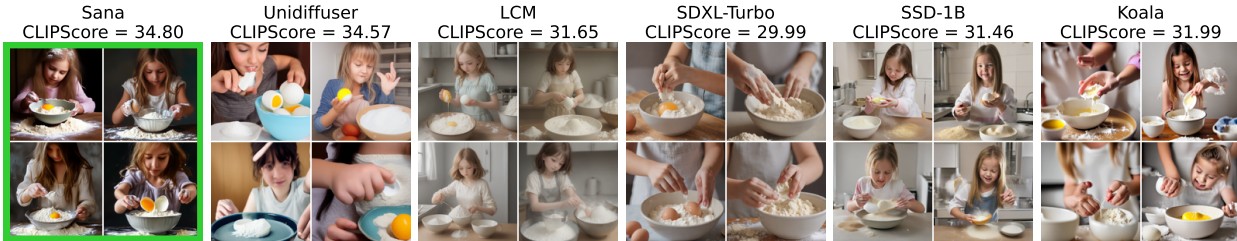

(b) Prompt: "*girl cracking an egg into a bowl of flour*"

Figure 1: Visual comparison of generated images and corresponding averaged CLIP scores from 6 different text-to-image models. All reported scores have a tolerance bounded by $\pm 0.5$ CLIPScore.

a prompt, selects a model, and receives feedback only for that selection. Specifically, it models expected rewards as linear functions in a kernelized prompt space and uses optimism-based exploration to guide model selection. From an outside perspective, this system can be compared to Mixture-of-Experts architectures, where features are routed towards the best-suited layers. The difference is that we consider here different models and we avoid any pretraining.

Although theoretically grounded, this approach suffers from several limitations in practice. First, it assumes a fixed kernel and a parametric reward structure. However, in real-world settings, the relationship between prompts and model performance can be highly nonlinear or irregular. Second, PAK-UCB disregards the structural or potential correlational relationships between models, despite the fact that numerous generative models exhibit architectural similarities or common training objectives. As a result, the algorithm struggles to leverage shared information across models. Empirically, we observe that PAK-UCB tends to generalize poorly when the number of models increases or when prompts become more diverse (see subsection 5.1), suggesting that these assumptions limit its scalability and adaptability.

To overcome these limitations, we introduce NeMoS, a novel approach designed for scalable and reliable generative model selection. Our method performs nonparametric reward estimation for each model using the CLIP embedding of the current prompt as well as historical reward observations from similar prompts. Unlike prior approaches, we enhance this learner with a limited active learning budget: at selected prompts, the algorithm can query the reward of all models for a given prompt. This additional signal is strategically used to resolve ambiguity, improve generalization across similar prompts, and uncover latent correlations between models. The challenge then becomes efficiently using the query budget based on the chosen metric, e.g., regret minimization.

By combining passive learning from partial feedback with targeted active querying, our proposed method successfully balances exploration and exploitation. Under mild assumptions on the smoothness of reward functions and the learnability of model behaviors, we derive a novel regret bound that formalizes the efficiency of our approach. These theoretical guarantees are corroborated by extensive experiments, which demonstrate that our method consistently outperforms state-of-the-art baselines. Importantly, we also show that even a small number of active queries can already yield substantial gains in selection accuracy and learning efficiency (see subsection 5.2).

**Our contributions are summarized as follows:**

- We propose NeMoS, a prompt-based online modeling selection algorithm for generative AI models, explicitly leveraging both prompt- and model-level similarities.

- The core of NeMoS is the non-parametric neighbor-based bandit combined with an active learning mechanism selectively querying on ambiguous and uncertain prompts. We estimate the ambiguity based on the gap between the estimated rewards, increasing the model confidence and accelerating convergence.

- We derive a regret bound showing our active, non-parametric approach improves over passive algorithms. We further confirm these gains empirically across six real-world text-to-image models (four prompt datasets) and LLM question answering tasks. Our approach outperforms state-of-the-art baselines under varying budgets, hyperparameters, and model pools.

## 2 Related works

**Offline approaches** learn the mapping from inputs to models using prompt-to-model ranking networks (Luo et al., 2024), dataset-to-model forecasting for fine-tuning decisions (Lewandowski et al., 2025), or prompt-specific leaderboard generation (Frick et al., 2025). When deployment and training distributions align, such predictors can work well, yet they do not re-calibrate in real time when prompts drift or when new models are introduced, which is what we consider in our setting. More adaptive strategies based on bandits initially focused on unconditional generators and therefore missed the prompt-specific nature of the problem (Hu et al., 2025a; Rezaei et al., 2025). PAK-UCB (Hu et al., 2025b) brings the contextual view we need by fitting, for each model, kernel ridge regression on CLIP embeddings and acting optimistically with UCB. In practice, however, fixing a kernel and estimating each model independently can underfit heterogeneous prompts and overlook correlations between models that share architectures or data. Our method answers these issues by remaining nonparametric, using neighborhoods instead of a global kernel, and by occasionally querying the full reward vector on the same prompt.

**Generated image evaluation** has evolved in parallel with this objective. Distribution level scores such as FID (Heusel et al., 2017) and Inception Score (Salimans et al., 2016) capture global realism and diversity, whereas other metrics focus on prompt alignment like CLIPScore (Hessel et al., 2021) or human preference datasets such as Pick-a-Pic (Kirstain et al., 2023) and HPSv2 (Wu et al., 2023). Recent surveys and leaderboards warn against single number verdicts and argue for multi dimensional protocols that cover relevance, realism, and diversity (Ku et al., 2024; Zhang et al., 2023). In this spirit, we report CLIPScore because it is widely adopted and correlates well with human preferences, while we frame our claims as relative improvements under any similar metric. This aligns precisely with the context in which an online selector is expected to deliver value.

**Active learning** (Settles, 2009; Hanneke, 2014) selects informative labels under a budget, for example with uncertainty sampling (Du et al., 2015) or query by committee and expected model change (Zhdanov, 2019). Recent works adapts this idea to sequential prediction by learning when to request extra feedback. For instance, Neuronal-s (Ban et al., 2024) uses two networks, a predictor for rewards and an auxiliary component for uncertainty, and triggers full feedback in a streaming setting through an uncertainty threshold. In our case we keep the estimator simple and nonparametric and use closed form bonuses rather than learned uncertainties.

## 3 Problem Definition

We consider a *contextual multi-armed bandit* setting where contexts lie in a metric space. Let $\mathcal{X}$ denote the context space, (i.e. prompt embeddings), equipped with a distance function $\rho : \mathcal{X} \times \mathcal{X} \to \mathbb{R}_{\geq 0}$, and let $\mu$ be a distribution over $\mathcal{X}$. We assume a finite set of $G$ models, denoted by $\mathcal{G}$, corresponding to generative models.

At each round $t = 1, 2, \ldots, T$, the learner observes a prompt $X_t \sim \mu$, and must choose a model $g_t \in \mathcal{G}$. For each model $g \in \mathcal{G}$, there exists an unknown reward function $f^g : \mathcal{X} \to [0, 1]$ such that the observed reward is

$$Y_t^g := f^g(X_t) + \eta_t^g, \tag{1}$$

where we assume that $\eta_t^g$ is a zero-mean sub-Gaussian noise (Assumption A.4). Only the reward (e.g. CLIPScore) $Y_t^{g_t}$ corresponding to the selected model is observed. In addition, we allow the learner to use a limited budget $B(T)$ of *queries*, where at selected rounds it can observe the rewards of all models on the current prompt to accelerate convergence.

Existing contextual bandit methods mainly differ in how they link the context to the expected reward. Parametric approaches such as LinUCB (Chu et al., 2011) or kernelized bandits (Valko et al., 2013; Hu et al., 2025b) assume a fixed functional form, for example linear or based on a kernel. They offer strong theoretical guarantees and learn quickly when the assumption is correct, but they tend to generalize poorly when prompts are diverse or when the number of models is large (see subsection 5.1). Non parametric approaches such as the Zooming Bandit (Slivkins, 2011) and KNN-UCB (Reeve et al., 2018) make fewer assumptions and are more flexible in heterogeneous settings, but they converge more slowly and their theoretical guarantees are weaker. A further limitation for our application is that most existing algorithms treat each model independently, while in practice generative models often share training data or architecture.

Our objective in this setting is to sequentially select models in order to minimize the cumulative regret over a time horizon of $T$ rounds. The cumulative regret $R(T)$ is defined as the sum of differences between the reward of the optimal model prompt-wise and the reward of the chosen model:

$$R(T) := \sum_{t=1}^{T} \left( f^{g_t^\star}(X_t) - f^{g_t}(X_t) \right), \tag{2}$$

where $g_t^\star = \arg\max_{g \in \mathcal{G}} f^g(X_t)$ denotes the optimal model for prompt $X_t$ and $g_t$ the model chosen by the algorithm. At each round $t$, the learner must select a model $g_t$ based solely on the history of past observations, without access to the reward vector except when a query is made.

## 4 The NeMoS algorithm

We now present our proposed method **NeMoS** in Algorithm 1, a contextual bandit algorithm tailored to prompt-based model selection. We build upon the $k$ Nearest Neighbors algorithm (Reeve et al., 2018) to combine non-parametric reward estimation with active learning under a limited query budget.

### 4.1 Nearest-neighbor reward estimation

At round $t$, we assign each model $g \in \mathcal{G}$ a score called a *UCB index* that combines a $k$-NN reward estimate at $X_t$ with an uncertainty bonus. The estimate is the average of the rewards of the $k$ nearest past observations of $g$, motivated by the assumption that similar prompts yield similar rewards. The bonus consists of a statistical term that decreases with $k$ and a geometric term that increases with the distance from $X_t$ to its $k$-th neighbor. We choose $k$ adaptively to balance these effects, and select the model with the largest index. For models without data, we assign an infinite index to ensure initial exploration.

**History and neighbors** Let $H_g(t) = \{(X_s, Y_s^g) : s < t$ and the reward of $g$ at $X_s$ was observed$\}$ be the history of model $g$ at timestep $t$ and $N_g(t) = |H_g(t)|$ its size. Given a candidate neighbor count $k \in \{1, \ldots, N_g(t)\}$, we denote by $\text{NN}_g(X_t, k) \subseteq H_g(t)$ the set of the $k$ nearest neighbors of $X_t$ in $H_g(t)$ under the metric $\rho$ (in practice, we set $\rho$ to the cosine distance in the CLIP embedding space), and let

$$r_{g,k}(t) = \max_{(x, \cdot) \in \text{NN}_g(X_t, k)} \rho(X_t, x)$$

be the distance to the $k$-th nearest neighbor.

**The UCB index**   We construct the UCB index $I_g(X_t)$ to approximate the reward of a given prompt $X_t$ for each model $g$. To do so, we first construct the $k$-NN reward estimate as the average over its $k$ closest past observations:

$$\hat{f}_g(X_t, k) \;=\; \frac{1}{k} \sum_{(x,y) \in \mathrm{NN}_g(X_t, k)} y. \tag{3}$$

We additionally define an uncertainty bonus on these observations to account for the uncertainty of our approximation. This term is constructed as a sum of two parts: (i) a *statistical uncertainty* term that shrinks with $k$ (proportionate to $k^{-1/2}$), and (ii) a *geometric uncertainty* term that grows with the neighbor radius $r_{g,k}(t)$:

$$U_g(X_t, k) \;=\; \sqrt{\frac{\theta \, \log N_g(t)}{k}} \;+\; \phi(t) \, r_{g,k}(t), \tag{4}$$

where $\theta > 0$ is a constant controlling exploration (cf. subsection B.1) and $\phi(t) > 0$ (non-decreasing) weights geometric uncertainty (in practice we set $\phi(t) = \log(t)$). The neighbor count chosen for the approximation is the value that *balances* these two sources of uncertainty:

$$k_g(t) \;=\; \underset{1 \leq k \leq N_g(t)}{\operatorname{argmin}} \; U_g(X_t, k). \tag{5}$$

We finally construct the UCB-index[1] using this $k_g(t)$, combining the reward estimate and the uncertainty bonus:

$$I_g(X_t) \;=\; \hat{f}_g\big(X_t, k_g(t)\big) \;+\; U_g\big(X_t, k_g(t)\big). \tag{6}$$

This approximation upper-bounds the true reward with high probability (see Appendix A).

**Model selection and cold start**   If $N_g(t) = 0$ (the model has never been used), we set $I_g(X_t) = +\infty$ to ensure initial exploration. The algorithm then plays

$$g_t \;=\; \underset{g \in \mathcal{G}}{\operatorname{argmax}} \; I_g(X_t). \tag{7}$$

## 4.2   Active Querying

We augment NeMoS with an *active learning* mechanism: at selected rounds, the algorithm may spend one query from its limited budget $B(T)$ to observe the rewards of *all* models on the current prompt $X_t$, rather than only the chosen arm. This additional feedback helps reduce ambiguity, accelerates the convergence of neighborhood estimates, and uncovers correlations between models. The central design choice is hence *when* to query. We propose the *Delta* rule to comply with our theoretical analysis and detail its construction thereafter. We also consider several alternative criteria as ablation study, but they consistently fail to perform as well as the main one empirically (see subsection 5.2). Related uncertainty-driven triggers have further been discussed in active learning surveys (Settles, 2009; Tharwat & Schenck, 2023).

**Primary criterion: *Delta* (top-two gap).**   Under this design, we trigger a query when the *gap between the top two UCB indices* at $X_t$ is small (below a threshold $\delta$). Intuitively, this criterion can be viewed as an adaptation of margin-based uncertainty sampling (Bahri et al., 2022) to the contextual bandit setting. Similar to how margin sampling targets instances near the decision boundary in classification, our strategy identifies rounds where the learner is "on the fence" between the two most promising candidates, suggesting that a full-feedback query is maximally informative there. This gap serves as a proxy for the difference between the future rewards: when it is large, the choice of the best model is essentially clear, but when it is small, the learner faces genuine ambiguity. By concentrating queries on these uncertain rounds, the algorithm gathers the most valuable information for refining neighborhood estimates, leading to the regret improvements formalized in Theorem 4.2.

---

[1]$I_g(X_t)$ is sometimes referred to as the Upper Confidence Bound (UCB) in the literature, but it should not be mistaken for the Uncertainty Bonus $U_g(X_t, k)$. To avoid confusion we prefer the use of the term *UCB-index* and notation $I_g$ for the former, and the term *Confidence Bonus* with the notation $U_g$ for the latter.

---

**Algorithm 1** NeMoS

---

1: **Input:** Horizon $T$, models $\mathcal{G}$, UCB parameter $\theta$, active query function **Q**, budget $B(T)$
2: Initialize $H_g(1) \leftarrow \emptyset$, $N_g(1) \leftarrow 0$ for all $g \in \mathcal{G}$
3: **for** $t = 1$ **to** $T$ **do**
4:     Observe new prompt $X_t$
5:     **for** each model $g \in \mathcal{G}$ **do**
6:       **if** $N_g(t) > 0$ **then**
7:         Compute $k_g(t)$ minimizing the UCB criterion
8:         Estimate reward $\hat{f}_g(X_t)$ over $k_g(t)$ neighbors
9:         Compute UCB index $I_g(X_t)$ & uncertainty $U_g(X_t)$
10:      **else**
11:         Set $I_g(X_t) \leftarrow +\infty$
12:     Select $g_t = \arg\max_g I_g(X_t)$
13:     **if** $\mathbf{Q}(X_t) = $ True and $B > 0$ **then**
14:       Query all rewards $\{Y_t^g\}$ and update all $H_g(t)$, $N_g(t)$
15:       Decrement budget $B \leftarrow B - 1$
16:     **else**
17:       Play $g_t$, observe $Y_t^{g_t}$, update $H_{g_t}(t)$, $N_{g_t}(t)$

---

**Alternatives.** We design three additional criteria to trigger the query of a full feedback on a given prompt:
*UCB-threshold*: query whenever the maximum uncertainty bonus $\max_{g \in \mathcal{G}} U_g(X_t)$ exceeds $\varepsilon$, which typically occurs in regions with few neighbors and higher variance.
*Warm-start*: query systematically during the first $B(T)$ rounds and then switch to passive KNN-UCB, effectively front-loading the budget.
*Variance-threshold*: for the selected arm $g_t$, query when the empirical variance of rewards among its $k_{g_t}(t)$ neighbors around $X_t$ exceeds a threshold $v$, indicating local inconsistency and the need for additional information.
Further details on these variants are provided in subsection C.2, and threshold selection and implementation details are explained in subsection B.2.

*Remark* 4.1. Unlike classical active learning, where queries usually request a single missing label, here one could also imagine partial queries that reveal the rewards of only a few models on each prompt. However, our experiments indicate that, for the same compute budget, full queries that reveal all model rewards consistently lead to better final performance (see Table 1 and Figure 2). We therefore focus on full queries in this work.

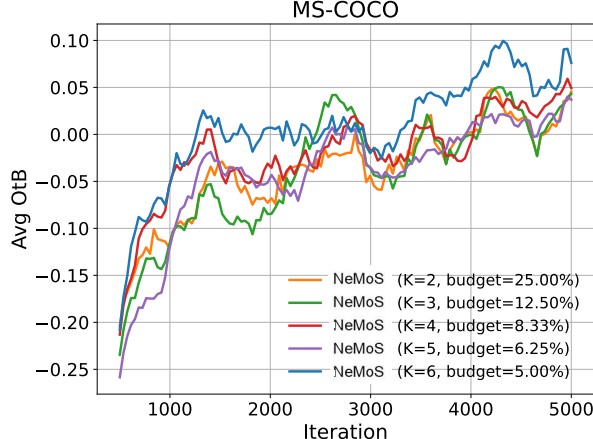

Figure 2: Sliding average OtB of NeMoS for different values of $K$ on MS-COCO.

| Number of models $(K)$ considered in the queries | Avg. regret |
|---|---|
| NeMoS ($K$=2, budget=25.00%) | $1.0505 \pm 0.012$ |
| NeMoS ($K$=3, budget=12.50%) | $1.0527 \pm 0.015$ |
| NeMoS ($K$=4, budget=8.33%) | $1.0362 \pm 0.009$ |
| NeMoS ($K$=5, budget=6.25%) | $1.0545 \pm 0.011$ |
| NeMoS ($K$=6, budget=5.00%) | $\mathbf{1.0085 \pm 0.007}$ |

Table 1: Average total regret of NeMoS for different values of $K$ on MS-COCO.

### 4.3 Theoretical Analysis

#### 4.3.1 Regret Guarantee

We provide a theoretical upper bound on the cumulative regret of NeMoS under the assumptions A.1–A.5, remaining consistent with previous works (cf. Reeve et al., 2018, Section 2.2). Full proof is provided in Appendix A. Intuitively, each active query uses one unit of the budget to observe all models' rewards on a carefully chosen prompt, immediately reducing uncertainty and improving all subsequent $k$-NN estimates in that region. However, full-feedback evaluations are costly, so we must keep the total number of queries sublinear in $T$. By setting $B(T) = T/\log T$, we ensure that queries are sufficiently frequent to drive the confidence bonuses, and hence the regret, down to a purely polylogarithmic rate, while still maintaining an overall budget that grows slower than the horizon.

**Theorem 4.2** (Regret bound under budgeted active querying)**.** *If the assumptions A.1–A.5 are verified, then, for $\theta > 2$ and active query budget $B(T) = T/\log T$, the cumulative regret of NeMoS after $T$ rounds satisfies:*

$$R(T) = \sum_{t=1}^{T} \left( f_{g_t^\star}(X_t) - f_{g_t}(X_t) \right) \leq C \log(T)^{\frac{d+2}{\alpha} + \frac{d+2}{2}} \tag{8}$$

*where $d$ is the intrinsic dimension of $\mathcal{X}$ (see Assumption A.1), $\alpha$ is the Tsybakov exponent (see Assumption A.3) and $C$ is a constant depending on $G$, $\theta$, $\alpha$, $\lambda$ (Lipschitz coefficient in Assumption A.2) and $d$.*

*Remark* 4.3. The use of active learning in NeMoS leads to a substantial improvement over the passive KNN-UCB baseline whose regret grows as $O\!\left(T^{1-\frac{\alpha+1}{d+2}}\right)$, which is close to linear. To the best of our knowledge, we are the first to obtain this regret bound in this setting.

#### 4.3.2 Time and Space Complexity of NeMoS

**Time complexity**  The overall time complexity of NeMoS over a horizon $T$ with $G$ models and query budget $B(T)$ is

$$O\!\left(G\,T^2 \log T + \left(T + G \cdot B(T)\right)I\right)$$

where $I$ is the maximum per-model inference cost. In practice, when using very large values of $T$, we can cap the per-arm history, reducing the empirical complexity to near-linear in T. Moreover, the $T^2$ term carries a very small constant: an empirical breakdown (Table 6 in the appendix) shows that the nearest-neighbor selection accounts for under 1% of NeMoS's GPU runtime at the horizons we consider, the rest being model inference.

**Proof of NeMoS Time complexity.**  We assume that each model $g \in \mathcal{G}$ has been played at least once by round $t$, which holds whenever $t > G$. Under this assumption, we have $N_g(t) > 0$ for all $g$, and at each round $t > G$, the algorithm performs three main operations:

1. Compute the distance between the current prompt $X_t$ and each entry in the history $H_g(t)$, an $O\!\left(N_g(t)\right)$ operation.

2. Sort these $N_g(t)$ distances to identify the nearest neighbors, which costs $O\!\left(N_g(t) \log\!\left(N_g(t)\right)\right)$.

3. Find the optimal number of neighbors $k_g(t)$ by minimizing the UCB term, which costs $O(N_g(t))$.

The per-arm cost over the horizon is therefore:

$$O\!\left(\sum_{t=1}^{T} N_g(t) \log\!\left(N_g(t)\right)\right) = O\!\left(\sum_{t=1}^{T} t \log t\right) = O\!\left(T^2 \log T\right).$$

In addition to this selection cost, we must account for the inference time of the models. Even though inference is a constant-time operation for a given model (with maximum cost $I$), it adds a total cost of $O\!\left((T + BG)I\right)$.

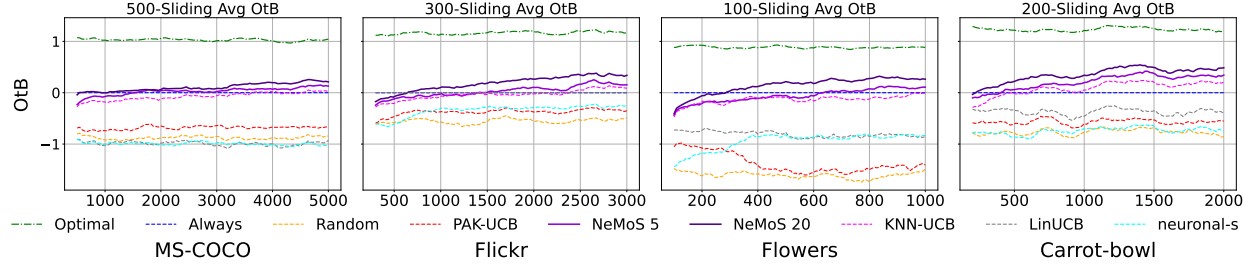

Figure 3: Sliding average OtB comparison between our algorithm and baselines across four prompt datasets with 6 models. Results are averaged over 10 runs. `NeMoS` outperforms all baselines and achieves a positive OtB on all datasets.

Summing both contributions, the overall time complexity becomes:

$$O\big(G\,T^2 \log T + (T + BG)I\big).$$

$\square$

**Space complexity.** At each iteration, we store the result for the selected arm $g_t$: a prompt embedding and its scalar reward. Additionally, at query steps, we store one such result for each arm $g \in \mathcal{G}$. As a result, the total number of stored entries is at most $T + BG$, where $B(T)$ is the budget of query steps. The overall memory complexity is therefore $O(T + BG)$.

## 5 Evaluation

We evaluate `NeMoS` by measuring the CLIPScore it achieves on different prompt datasets, and compare this performance with standard bandit baselines to assess both the overall generation quality and the accuracy of model selection.

**Models.** Throughout the evaluations, we use six different text-to-image models: Sana 1.5 (Xie et al., 2024), LCM Dreamshaper v7(Luo et al., 2023), Unidiffuser v1(Bao et al., 2023), SDXL-Turbo(Podell et al., 2024), SSD-1B(Gupta et al., 2024), and Koala-Lightning-700M(Lee et al., 2024). All models are accessed through the `diffusers`[2] library and executed with appropriate settings for resolution and number of inference steps (see Table 4 in the appendix).

**Prompt datasets.** We evaluate our method on four different prompt datasets, all accessible via the Hugging Face `datasets` library[3]. The first two, MS-COCO (Lin et al., 2014) and Flickr30k (Plummer et al., 2015), are broad and diverse, making the model selection task more challenging due to the strong heterogeneity of the prompts. The remaining two are more focused: one is a flower caption dataset[4], and the other is a subset that we manually extracted from MS-COCO of pictures using the carrot & bowl tag (4640 prompts). These more constrained domains allow us to highlight the effectiveness of our algorithm in low-variance settings and confirm that model selection becomes increasingly difficult as prompt diversity increases.

**Metrics.** We evaluate the performance of the algorithms using two main metrics. The first is *Outscore-to-Best (OtB)*, defined as the average difference between the CLIPScore obtained by the algorithm and that of the single best model across the dataset. In the experiments, we report the sliding average OtB, which is computed by averaging the OtB values over a fixed-size window of recent iterations. This smooths the curves and highlights the overall performance trend, rather than the fluctuations at individual rounds. The second metric is the *Optimal Pick Ratio (OPR)*, which measures the proportion of times the algorithm selects the best model for a given prompt. Together, these metrics capture both the absolute quality of the selected generations and the algorithm's ability to identify the prompt-specific optimal model.

---

[2]`https://huggingface.co/docs/diffusers/index`
[3]`https://huggingface.co/datasets`
[4]`https://huggingface.co/datasets/pranked03/flowers-blip-captions`

**Baselines.** We compare the **Delta** variant of NeMoS with budgets of 0, 5 and 20% of the horizon to several standard baselines from the contextual bandit literature, including **LinUCB** (Chu et al., 2011), **PAK-UCB** (Hu et al., 2025b), and an active bandit baseline: **neuronal-s** (Ban et al., 2024) (with a budget of 20% of $T$). In addition, we include three reference baselines in our evaluation plots: a random selection strategy (**random**), an oracle that always selects the best model for each prompt (**optimal**), and a static baseline that always selects the same model: the model that has the maximum average CLIPScore over the whole dataset (**always**).

*Remark* 5.1. To ensure a fair and meaningful comparison between active and passive algorithms, we always select the model $g_t$ *before* issuing a query, based solely on past observations. Even when the algorithm decides to query the full reward vector, this additional information is only used to update the history and guide future choices, not to select the optimal model at the current round. This ensures that any performance differences truly reflect the benefit of improved information acquisition over time, rather than being driven by immediate access to ground-truth rewards during query rounds. Results when using the query to guide selection are reported in the appendix.

## 5.1 Results overview

Figure 3 shows the OtB performance of our algorithm compared to the baselines across the four prompt datasets, using a pool of six generative models. The corresponding budget consumption and OPR curves are reported in the appendix. Our method consistently outperforms all baselines (see Table 2 for numerical values), including the passive version of NeMoS without active queries. This highlights the benefit of incorporating an active learning strategy into the selection process. Even with a very limited query budget of only 5% of the horizon $T$, our algorithm achieves significant performance gains, showing that a small number of strategically placed full-feedback queries can substantially improve learning.

Interestingly, NeMoS 5 improves over the Always baseline while requiring lower total GPU cost (see Table 5 in the appendix ). This shows that NeMoS successfully leverages smaller models when beneficial for specific prompts.

Table 2: **Average cumulative regret comparison.** We report the mean regret $\pm$ standard deviation over 10 independent runs. NeMoS consistently achieves the lowest regret across all datasets.

| Algorithm | MS-COCO | Flickr | Flowers | Carrot-bowl |
|---|---|---|---|---|
| Optimal | 0.000 ±0.000 | 0.000 ±0.000 | 0.000 ±0.000 | 0.000 ±0.000 |
| Always | 1.032 ±0.015 | 1.161 ±0.018 | 0.884 ±0.012 | 1.232 ±0.014 |
| Random | 1.905 ±0.084 | 1.713 ±0.076 | 2.476 ±0.112 | 2.003 ±0.095 |
| Neuronal-s | 2.023 ±0.145 | 1.511 ±0.120 | 1.845 ±0.138 | 1.976 ±0.152 |
| PAK-UCB | 1.714 ±0.092 | 1.546 ±0.085 | 2.243 ±0.105 | 1.800 ±0.098 |
| LinUCB | 2.013 ±0.065 | 1.161 ±0.042 | 1.690 ±0.058 | 1.603 ±0.061 |
| KNN-UCB | 1.112 ±0.031 | 1.206 ±0.029 | 1.031 ±0.025 | 1.158 ±0.033 |
| **NeMoS** (5%) | 1.016 ±0.021 | 1.141 ±0.024 | 0.953 ±0.019 | 1.032 ±0.022 |
| **NeMoS** (20%) | **0.930 ±0.018** | **0.989 ±0.020** | **0.767 ±0.015** | **0.894 ±0.017** |

As further shown in Table 2, NeMoS substantially improves over existing methods: it reduces the average regret of PAK-UCB by roughly 40-60% across datasets, and introducing a 20% query budget lowers the regret by an additional 15-25% compared to the passive KNN-UCB baseline. These gains stem from two key aspects of our approach: first by averaging over neighboring prompts in the embedding space, NeMoS can generalize feedback beyond individual samples, which is particularly effective in more homogeneous domains such as Flowers or Carrot-Bowl; and second by issuing active queries in ambiguous regions, the algorithm reduces wasted exploration on clearly suboptimal models. Together, these mechanisms enable NeMoS to converge faster and to achieve a higher OPR, on average 10 percentage points better than all baselines as we see in Figure 22 in the appendix

The improvements in total regret are most pronounced on the less diverse datasets, which is consistent with our theoretical analysis. These datasets exhibit lower values of the ratio $\frac{d+2}{\alpha}$ (see Table 10 in the appendix), which appears in the regret bound Theorem 4.2, indicating that nearest-neighbor estimations are more accurate. This corroborates our heuristic: the more homogeneous the prompts, the easier it is to distinguish between the performance of different models.

Moreover, our algorithm surpasses the performance of the single best model on each dataset, demonstrating its ability to leverage the complementarity among available models and dynamically adapt to different prompt types.

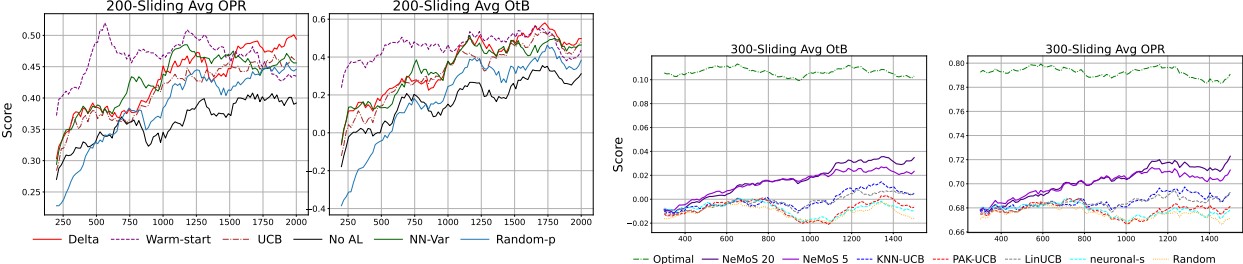

Figure 4: Performance of different uncertainty estimation methods on the carrot-bowl dataset. Results are averaged over 20 runs. OPR (on the left) and OtB (on the right) are reported. The **Delta** variant achieves the best final OtB and OPR values.

Figure 5: Performance evolution of NeMoS and baseline methods on the CommonsenseQA dataset, using two language models. Metrics shown are OtB and OPR over time. Results are averaged over 10 runs.

## 5.2 Active learning metrics

Figure 4 compares the performance of five query trigger strategies against a passive baseline. The **Delta** strategy consistently attains the best final OtB and OPR because it concentrates queries where they are most informative: rounds in which the top two candidates are nearly tied. By injecting full feedback precisely at these decision boundaries, NeMoS resolves model ambiguity early and reduces downstream model-switch errors. In contrast, **UCB-threshold** tends to fire in globally uncertain (sparsely sampled) regions regardless of competitiveness between models, which can diffuse the budget. **Variance-threshold** is more reactive to local noise and may over-query in heterogeneous neighborhoods. **Warm-start** front-loads the budget, yielding a fast initial lift but slower gains later once the query supply is exhausted. Lastly, a **Random Queries** baseline triggers full-feedback uniformly at random with probability $B(T)/T$, providing a sanity check that improvements stem not only from having more labels, but also from querying at the right rounds.

Finally, Table 2 shows that a small budget of 5% already delivers a sizeable gain over the passive baseline (10-15% lower regret across datasets), meaning that limited access to ground-truth feedback is enough to calibrate neighborhoods and sharpen model comparisons. Increasing the budget beyond 20% yields diminishing returns: once decision boundaries are well resolved and neighbor distances shrink, additional queries rarely change the selected model, so performance plateaus.

## 5.3 Generalization to Language Modeling

In this section, we present experiments where the text-to-image (T2I) task is replaced by a language modeling task. Input prompts are sampled from the CommonsenseQA dataset Talmor et al. (2019), and we consider two LLMs: Gemma 2B Gemma Team et al. (2024) and LLaMA 3B Grattafiori et al. (2024). In this setting, the reward is binary, with a value of 1 if the selected model provides a correct answer, and 0 otherwise. The input of all baselines is the RoBERTa embeddings of the prompts Liu et al. (2019). Performance is evaluated using OtB and OPR metrics, as shown in Figure 5. NeMoS is able to very well adapt to this different task, by achieving a positive OtB and the best OPR among all baselines, even with a budget of only 5% of $T$. These results support the generalization of NeMoS to broader generative model selection tasks

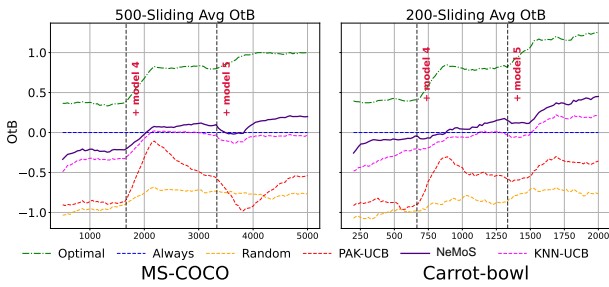 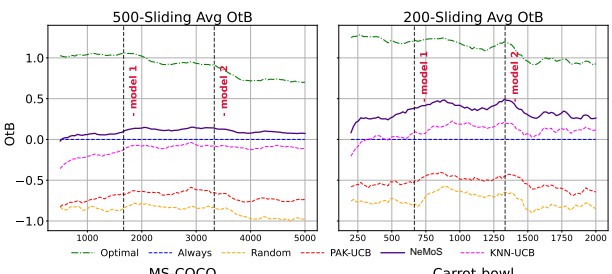

Figure 6: Performance of NeMoS with a 20% budget and other baselines under the model addition setup. OtB is reported. Results are averaged over 10 runs. NeMoS adapts to both strong and weak models by allocating its budget strategically.

Figure 7: Performance of NeMoS with a 20% budget and other baselines under the model removal setup on 2 datasets (MS-COCO on the left and Carrot-bowl on the right). OtB is reported. Results are averaged over 10 runs.

beyond the text-to-image setting. Further work should explore the specifics of an efficient implementation in this use-case, notably with regard to baselines designed specifically for this purpose.

### 5.4 Adaptability and Sensitivity Analysis

#### 5.4.1 Model addition

Given the rapid evolution of state-of-the-art architectures, a practical model selection framework must be robust to the frequent introduction or removal of candidate models. To evaluate the adaptability of our algorithm in such non-stationary settings, we consider a dynamic setup where new models are added during the evaluation phase. Initially, the model pool contains only three generators (Unidiffuser , LCM and SSD-1B). At time step $^1/_3 T$, SDXL-Turbo (Podell et al., 2024) is introduced, followed by Sana (Xie et al., 2024) at time step $^2/_3 T$. Results are reported in Figure 6.

This experimental design allows us to test the algorithm's responsiveness to changes in the available action space. We observe that our method quickly adapts to the addition of new models, whether the newly added model is highly efficient or relatively weak. In both cases, the algorithm efficiently explores and integrates the new options into its decision-making process, adjusting its selection strategy accordingly. Specifically, on the Carrot-Bowl dataset, NeMoS achieves a 21% lower total regret compared to KNN-UCB, and 48% lower than PAK-UCB in this dynamic setting. This robustness to evolving model pools further highlights the practical value of our approach in real-world scenarios where new models may be introduced or deleted over time.

#### 5.4.2 Model removal

To further assess adaptability, we also investigate a complementary model removal scenario where the pool of available generators is progressively reduced during evaluation. Starting with all six models, Unidiffuser is removed at time step $^1/_3 T$, followed by SSD-1B at $^2/_3 T$. Results, reported in Figure 7, show that NeMoS remains robust to such contractions of the action space. The algorithm efficiently reallocates its budget toward the remaining candidates, maintaining competitive performance despite the reduced diversity. On MS-COCO and Carrot-Bowl, NeMoS consistently outperforms baseline strategies, sustaining lower regret even after strong model removals, which highlights its resilience to real-world settings where underperforming or costly models may be discarded.

#### 5.4.3 Ablation studies

We conduct several additional ablations to further assess the robustness of NeMoS across architectural and algorithmic choices, with the corresponding results detailed in subsection C.7. Replacing CLIP with BERT (Devlin et al., 2019) textual embeddings leads to comparable trends, showing that NeMoS does not

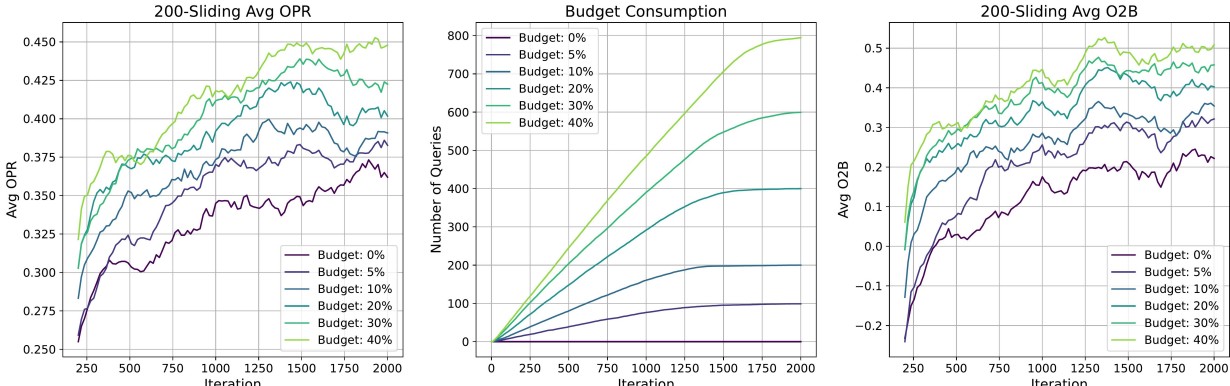

Figure 8: Performance of NeMoS with different budget on the carrot-bowl dataset with 6 models. OPR (on the left), budget consumption (in the middle) and OtB (on the right) are reported. Results are averaged over 20 runs.

rely on a specific text encoder. Using a fixed neighborhood size instead of an adaptive value consistently worsens performance, which confirms the importance of dynamically choosing the number of neighbor $k$. When changing the reward metric to ImageReward (Xu et al., 2023) instead of CLIPScore, all performance rankings remain unchanged, indicating that our conclusions are not dependent on a particular reward metric. We further confirm this with a third, learned human-preference metric, HPSv2 (Wu et al., 2023), on which NeMoS again attains the lowest regret (see subsection C.8). Testing an alternative geometric uncertainty schedule with $\phi(t) = \sqrt{\log t}$ yields only small variations, suggesting that the confidence design is stable across reasonable choices. Moreover, sensitivity analyses on the Delta threshold and the exploration parameter $\theta$ (see subsection B.1) demonstrate that NeMoS maintains consistent performance across a broad range of values, confirming its robustness to hyperparameter settings.

We study the performance of NeMoS under different budget constraints and report the results in Figure 8. As expected, we observe larger budgets yield monotonically better accuracy. It appears that the marginal improvement in accuracy decreases as the budget grows, suggesting that the algorithm successfully identifies the prompts for which querying the ground truth is more useful. In every setting, the budget consumption is spread over the entire horizon, enabling stable performance.

In a last experiment, we report the average estimation error over all models, comparing NeMoS 20 to KNN-UCB. More precisely, the plotted quantity is the error between the true reward of the models: $Y_t^g$, and the algorithms reward estimate: $f_g(X_t, k_g(t))$, averaged over all models, i.e.: $E(t) = \frac{1}{G} \sum_{g=1}^{G} |Y_t^g - \hat{f}_g(X_t, k_g(t))|$. The results show that our active querying strategy leads to a faster reduction in estimation error, confirming that early and well positioned queries accelerate the convergence of the nearest neighbor estimates, explaining the efficiency of our active learning component.

# 6  Conclusion

We present NeMoS, an online framework for prompt-wise generative model selection. Unlike existing contextual bandit approaches that converge slowly and overlook semantic relationships between prompts, our method exploits similarities across prompts through a non-parametric, neighbor-based bandit design, and integrates an active learning component that queries ground-truth rankings only when they are most informative. This combination directly addresses the key limitations of prior work, enabling faster convergence and better generalization. We theoretically derive a polylogarithmic regret bound, demonstrating the tightness of our confidence bounds and formalizing the benefit of selective querying. Empirically, we conduct extensive evaluations on four datasets with six generative models, showing that NeMoS consistently outperforms both state-of-the-art bandit baselines and individual models, with regret reductions of up to 60%. A promising direction for future work is to design a cost-aware extension of NeMoS that explicitly accounts for the hetero-

geneous inference costs of different models, enabling the algorithm to balance predictive performance with computational efficiency in practical deployments.

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

# Appendix

## Table of Contents

# A  The regret bound

## A.1  Assumptions

We make the following assumptions throughout our analysis (these are the same as in Reeve et al. (2018)):

**Assumption A.1.** *(Intrinsic dimension).  There exist constants $C_d > 0$, $d > 0$, and $R_{\mathcal{X}} > 0$ such that for all $x \in \text{supp}(\mu)$ and all $r \in (0, R_{\mathcal{X}})$, we have:*

$$\mu(B(x, r)) \geq C_d \cdot r^d, \tag{9}$$

*where $B(x, r)$ denotes the open ball of radius $r$ centered at $x$ under the distance $\rho$.*

This assumption ensures that the distribution of prompts $\mu$ is sufficiently regular and that $\mathcal{X}$ locally behaves like a $d$-dimensional manifold.

**Assumption A.2.** *(Lipschitz continuity)  There exists a constant $\lambda > 0$ such that for all models $g \in \mathcal{G}$ and all $x, x' \in \mathcal{X}$,*

$$|f_g(x) - f_g(x')| \leq \lambda \cdot \rho(x, x'). \tag{10}$$

This assumption states that similar prompts should yield similar expected rewards for a given model (i.e., smoothness of the reward functions $f_g$). We stress that the Lipschitz condition constrains the *expected* score $f_g(x) = \mathbb{E}[Y_t^g \mid X_t = x]$, and not the realized score $Y_t^g = f_g(x) + \eta_t^g$. Only the mean-reward surface is required to be smooth; the per-image observations $Y_t^g$ may still fluctuate sharply between nearby prompts through the sub-Gaussian noise $\eta_t^g$ (Assumption A.4). Hence apparent jumps in raw CLIPScores between close prompts do not contradict Assumption A.2.

**Assumption A.3.** *(Tsybakov Margin)  Let $\Delta_g(x) = f^*(x) - f_g(x)$ where $f^*(x) = \max_{a \in \mathcal{G}} f_a(x)$, and define $\Delta(x) = \min\{\Delta_g(x) : \Delta_g(x) > 0\}$ if the minimum exists, and $0$ otherwise. We assume there exist constants $C_\alpha > 0$, $\delta_\alpha > 0$, and $\alpha > 0$ such that for all $\delta \in (0, \delta_\alpha)$,*

$$\mu(\{x \in \mathcal{X} : 0 < \Delta(x) < \delta\}) \leq C_\alpha \cdot \delta^\alpha. \tag{11}$$

This margin condition quantifies the diffiulty of the model selection problem: it controls the measure of prompts for which several models are nearly optimal. We verify this condition empirically in subsection C.6: on real prompt distributions the near-tie mass $\mu(\{0 < \Delta(x) < \delta\})$ is well described by $C_\alpha \delta^\alpha$ with a strictly positive exponent ($\alpha \approx 0.78$ on Carrot-bowl and $\alpha \approx 0.83$ on Flowers), confirming that the low-noise condition holds in practice.

**Assumption A.4.** *(Sub-Gaussian noise)  For each model $g \in \mathcal{G}$ and time $t$, the reward noise is conditionally sub-Gaussian: for all $x \in \mathcal{X}$ and $\eta \in \mathbb{R}$,*

$$\mathbb{E}\left[\exp\left(\eta \cdot (Y_t^g - f_g(x))\right) \mid X_t = x\right] \leq \exp\left(\frac{\eta^2}{2}\right). \tag{12}$$

**Assumption A.5.** *(Bounded rewards)  For all $t$ and $g \in \mathcal{G}$, we have:*

$$Y_t^g \in [0, 1]. \tag{13}$$

## A.2  Notations

We recap all the notations used throughout the paper in this table.

Table 3: Notations

| Notation | Description |
| --- | --- |
| $\mathcal{G}$ | Set of generative models (arms) |
| $G$ | Number of models |
| $T$ | Horizon (number of rounds) |
| $X_t$ | Prompt observed at round $t$ |
| $g_t$ | Model selected at round $t$ |
| $g_t^\star$ | Optimal model for $X_t$ (argmax over $g \in \mathcal{G}$) |
| $Y_t^g$ | Observed reward of model $g$ at round $t$ |
| $H_g(t)$ | History of observed pairs for model $g$ up to round $t$ |
| $N_g(t)$ | Size of $H_g(t)$ (number of feedback points up to $t$) |
| $\rho(x, x')$ | Distance (metric) between prompts $x$ and $x'$ |
| $\mathrm{NN}_g(X_t, k)$ | $k$ nearest neighbors of $X_t$ in $H_g(t)$ |
| $r_{g,k}(t)$ | Neighbor radius (distance to the $k$-th nearest neighbor) |
| $k_g(t)$ | Chosen number of neighbors at round $t$ |
| $\hat{f}_g(X_t, k)$ | $k$-NN reward estimate for model $g$ at $X_t$ |
| $U_g(X_t, k)$ | Confidence bonus (statistical + geometric) |
| $I_g(X_t)$ | UCB index of model $g$ at $X_t$ |
| $\theta, \ \phi(t)$ | Exploration parameter; geometric-uncertainty weight (non-decreasing) |
| $B(T)$ | Active-learning query budget as a function of $T$ |
| $Q(X_t)$ | Query trigger predicate at prompt $X_t$ |
| $\delta, \ \varepsilon, \ v$ | Thresholds for Delta, UCB-threshold, Variance-threshold |
| $\hat{f}_{(1)}(X_t), \ \hat{f}_{(2)}(X_t)$ | Largest / second-largest estimated rewards across models |
| $\widehat{\Delta}(X_t)$ | Top-two gap (largest minus second-largest estimate) |
| $R(T)$ | Cumulative regret up to time $T$ |
| $d, \ \lambda, \ \alpha, \ C$ | Intrinsic dimension; Lipschitz constant; Tsybakov exponent; theorem constant |
| $I$ | Per-model inference cost (time-complexity analysis) |

### A.3 Proof of Theorem 4.2

In this section we prove the regret bound for the Delta-variant of NeMoS, since it is the one we use in all the experiments, and the best one empirically.

For a subset $S \subset X$ and a model $g \in \mathcal{G}$, we define the **minimum gap of model $g$ in region $S$** as:

$$\Delta_g(S) := \inf_{x \in S} \Delta_g(x). \tag{14}$$

and $\Delta(S) = \min_{g \in \mathcal{G}} \Delta_g(S)$. Note that $\Delta(x)$ refers to the true gap function, and should not be confused with its empirical estimate $\hat{\Delta}(x)$ computed from finite samples.

We split the cumulative regret according to the "good event" $V(t)$ on which all UCB indices are valid:

$$V_g(t) = \{\hat{f}_g(X_t, k_g(t)) - U_g(X_t, k_g(t)) \le f_g(X_t) \le \hat{f}_g(X_t, k_g(t)) + U_g(X_t, k_g(t))\} \tag{15}$$

$$V(t) = \bigcap_{g \in \mathcal{G}} V_g(t) \tag{16}$$

Let $r(t) = f_{g_t^\star}(X_t) - f_{g_t}(X_t)$ so that $R(T) = \sum_{t=1}^{T} r(t)$, we can then split the regret in two terms :

$$r(t) = r(t) \, \mathbb{1}_{V(t)} + r(t) \, \mathbb{1}_{V(t)^\complement}. \tag{17}$$

The following lemma bounds the number of times we pull a suboptimal model in a given region $S$.

**Lemma A.6.** *Let $S \subset X$, and consider a model $g \in \mathcal{G}$. On the event $V(T)$, if $\Delta_g(S) > 2\phi(T) \cdot \mathrm{diam}(S)$, then the number of times model $g$ is selected in region $S$ while being suboptimal satisfies*

$$N_T^g(S) \leq \frac{4\theta \log T}{(\Delta_g(S) - 2\phi(T) \cdot \mathrm{diam}(S))^2} + 1. \tag{18}$$

*Proof.* Assume without loss of generality that $N_T^g(S) > 1$. Let $t$ be the last round where model $g$ is selected and the context $X_t$ lies in $S$, and where the event $V(t)$ holds:

$$t := \max \{s \leq T \mid X_s \in S, \ g_s = g, \ V(s) \text{ holds}\}. \tag{19}$$

Let $k(S)$ be the number of neighbors used to estimate $\widehat{f}_g(X_t, k_g(t))$ that are also in $S$:

$$k(S) := |\{(X, r) \in \mathrm{NN}_g(X_t, k_g(t)) \mid X \in S\}|. \tag{20}$$

Then, all neighbors used to compute $\widehat{f}_g(X_t, k_g(t))$ lie within $S$, and their distances to $X_t$ are at most $\mathrm{diam}(S)$. Since each new selection of $g$ in $S$ adds a new point to its history within $S$, we have:

$$N_T^g(S) \leq k(S) + 1. \tag{21}$$

Now, let $g_t^\star$ denote the optimal model at $X_t$, so $f_{g_t^\star}(X_t) = \max_{g'} f_{g'}(X_t)$. Since $g$ is selected at round $t$:

$$I_{g_t^\star}(X_t) \leq I_g(X_t), \tag{22}$$

where $I_g(X_t) = \widehat{f}_g(X_t, k_g(t)) + U_g(X_t, k_g(t))$. Using the good event $V(t)$, we also know:

$$f_{g_t^\star}(X_t) \leq I_{g_t^\star}(X_t), \tag{23}$$
$$f_g(X_t) \geq I_g(X_t) - 2U_g(X_t, k_g(t)). \tag{24}$$

Subtracting:

$$f_{g_t^\star}(X_t) - f_g(X_t) \leq 2U_g(X_t, k_g(t)). \tag{25}$$

But $f_{g_t^\star}(X_t) - f_g(X_t) = \Delta_g(X_t) \geq \Delta_g(S)$, so:

$$\Delta_g(S) \leq 2U_g(X_t, k_g(t)). \tag{26}$$

Now use the definition of $U_g$:

$$U_g(X_t, k_g(t)) = \sqrt{\frac{\theta \log T}{k(S)}} + \phi(T) \cdot \mathrm{diam}(S). \tag{27}$$

Hence:

$$\Delta_g(S) \leq 2 \left( \sqrt{\frac{\theta \log T}{k(S)}} + \phi(T) \cdot \mathrm{diam}(S) \right). \tag{28}$$

Solving for $k(S)$ yields:

$$k(S) \leq \frac{4\theta \log T}{(\Delta_g(S) - 2\phi(T) \cdot \mathrm{diam}(S))^2}. \tag{29}$$

Thus:

$$N_T^g(S) \leq k(S) + 1 \leq \frac{4\theta \log T}{(\Delta_g(S) - 2\phi(T) \cdot \mathrm{diam}(S))^2} + 1, \tag{30}$$

which concludes the proof. $\qquad\square$

Since the rewards are between 0 and 1, the total regret is bounded by the number of times the algorithm pulls suboptimal models. Then, we deduce from this the total regret over a region $S \subset X$ with $\Delta_g(S) > 2\phi(T) \cdot \text{diam}(S)$:

$$R_V(S,T) := \sum_{t=1}^{T} r(t) \mathbf{1}_{V(t)} \mathbf{1}_{X_t \in S} \leq G \left( \frac{4\theta \log T}{(\Delta(S) - 2\phi(T) \cdot \text{diam}(S))^2} + 1 \right). \tag{31}$$

The following lemma shows that for $t$ large enough, our estimate of $\Delta$ is close to its true value.

**Lemma A.7.** *Let $t \leq T$, and $N(t) = \min_{g \in \mathcal{G}} N_g(t)$*

$$\left| \Delta(X_t) - \widehat{\Delta}(X_t) \right| \leq 2 \frac{\sqrt{\theta \log T} + \phi(t)}{N(t)^{\frac{1}{d+2}}} \tag{32}$$

*Proof.* Let $t \leq T$. We have:

$$\frac{\left| \Delta(X_t) - \widehat{\Delta}(X_t) \right|}{2} \leq \max_{g \in \mathcal{G}} U_g(X_t, k_g(t))$$

With:

$$U_g(X_t, k) = \sqrt{\frac{\theta \log N_g(t)}{k}} + \phi(t) \cdot \max_{(x,\cdot) \in \text{NN}_g(X_t,k)} \rho(X_t, x)$$

Let $g \in \mathcal{G}$ and let us choose $k'_g = N_g(t)^{\frac{2}{d+2}}$. Since $k_g(t)$ minimizes the confidence bonus of model $g$, we know that:

$$U_g(X_t, k_g(t)) \leq \sqrt{\frac{\theta \log N_g(t)}{k'_g}} + \phi(t) \cdot \max_{(x,\cdot) \in \text{NN}_g(X_t,k'_g)} \rho(X_t, x) \leq \frac{\sqrt{\theta \log(T)}}{N_g(t)^{\frac{1}{d+2}}} + \frac{\phi(T)}{N_g(t)^{\frac{1}{d+2}}}$$

Where the inequality about the distance to the $k'_g$-th nearest neighbor in a space of dimension $d$ with $N_g(t)$ points:

$$\max_{(x,\cdot) \in \text{NN}_g(X_t,k'_g)} \rho(X_t, x) \sim \left( \frac{k'_g}{N_g(t)} \right)^{\frac{1}{d}}$$

is proven in Bhattacharyya & Chakrabarti (2008, Section II). Taking the maximum over all models $g \in \mathcal{G}$ proves the lemma.

$\square$

Let $N_0 := \left( \frac{4}{\varepsilon} \left( \sqrt{\theta \log(T)} + \phi(T) \right) \right)^{d+2}$, and let us assume that every model $g$ has a total number of play $N_g(T) \geq N_0$. If that's not the case for some models $g$, then these models contribute at most $N_0$ to the regret. Let $T_0$ be such that $\forall t \geq T_0$ we have $N(t) \geq N_0$. Thus, by Lemma A.7 and the definition of $T_0$, we have $\left| \Delta(X_t) - \widehat{\Delta}(X_t) \right| \leq \frac{1}{2}\varepsilon$ and therefore :

$$\widehat{\Delta}(X_t) \geq \varepsilon \implies \Delta(X_t) \geq \frac{\varepsilon}{2} \tag{33}$$

We can now decompose the total regret:

$$R(T) = \sum_{t=1}^{T} \mathbb{1}_{t \leq T_0} \mathbb{1}_{V(t)} r(t) + \sum_{t=1}^{T} \mathbb{1}_{t \geq T_0} \mathbb{1}_{V(t)} r(t) + \sum_{t=1}^{T} \mathbb{1}_{V(t)^{\complement}} r(t) \tag{34}$$

with the first sum :

$$\sum_{t=1}^{T} \mathbb{1}_{t \leq T_0} \mathbb{1}_{V(t)} r(t) \leq G T_0$$

In order to bound the second sum, we now have to partition the space $X$ into regions that each satisfy the condition $\varepsilon > 4\phi(T) \cdot \text{diam}(S)$ and to use Lemma A.6 on these regions, because we know that for every prompt $X_t$ in this sum, $\Delta(X_t) \geq \frac{\varepsilon}{2}$, or else the algorithm would have chosen to query (according to Equation 33). This can be done in a straightforward manner, and the number of such regions is therefore upper bounded by $O\left(\left(\frac{1}{\varepsilon}\phi(T)\right)^d\right)$, where $d$ is the intrinsic dimension of the space (see Vershynin, 2012, Lemma 5.2). On all of these regions, the regret is bounded by Lemma A.6 by:

$$R(S, T) \leq G \left( \frac{4\theta \log T}{\left(\frac{\varepsilon}{2}\right)^2} + 1 \right). \tag{35}$$

The second sum is then bounded by this regret multiplied by the number of regions :

$$\sum_{t=1}^{T} \mathbb{1}_{t \geq T_0} \mathbb{1}_{V(t)} r(t) \leq 4\theta G \left( \frac{\log T}{\varepsilon^2} \left( \frac{\phi(T)}{\varepsilon} \right)^d \right). \tag{36}$$

By the Tsybakov assumption, the budget needed for our active algorithm is:

$$T\mu(0 \leq \Delta(X) \leq \varepsilon) = TC_\alpha \varepsilon^\alpha \tag{37}$$

Then, $\varepsilon = \left( \frac{B}{C_\alpha T} \right)^{\frac{1}{\alpha}}$, so we can bound the regret by a function of the budget :

$$R(T) \leq 4\theta G \cdot \log T \cdot \phi(T)^d \left( \frac{C_\alpha T}{B} \right)^{\frac{2+d}{\alpha}} + G \left( 4 \left( \frac{C_\alpha T}{B} \right)^{\frac{1}{\alpha}} \left( \sqrt{\theta \log(T)} + \phi(T) \right) \right)^{d+2} \tag{38}$$

We now set the query budget as a function of the time horizon:

$$B(T) = \frac{T}{\log T}.$$

Plugging this into the regret bound gives the following expression:

$$R(T) \leq 4\theta G \cdot \log T \cdot \phi(T)^d \cdot (C_\alpha \log T)^{\frac{2+d}{\alpha}} + G \cdot \left( 4 (C_\alpha \log T)^{\frac{1}{\alpha}} \left( \sqrt{\theta \log T} + \phi(T) \right) \right)^{d+2}. \tag{39}$$

We now extract the leading term in $\log T$ to express the regret asymptotically. Ignoring constant factors and lower-order terms (which can be taken into account by the constant $C$), and taking $\phi(t) = \lambda$ we obtain:

$$R(T) \leq C \cdot \log(T)^{\frac{d+2}{\alpha} + \frac{d+2}{2}},$$

where $C$ is a constant that depends on $G$, $\theta$, $\alpha$, $C_\alpha$, $\lambda$ and $d$.

**Lemma A.8.** *The contribution to the regret from the iterations $t$ for which $V(t)$ is not true (i.e. the third sum in Equation 34) is a constant:*

$$\sum_{t=1}^{T} \mathbb{E} \left[ \mathbb{1}_{V(t)^c} \right] = O(1). \tag{40}$$

*Proof.* Recall that the event $V(t)$ holds if, for all models $g \in \mathcal{G}$, the following inequality is satisfied:

$$\left| \widehat{f}_g(X_t, k_g(t)) - f_g(X_t) \right| \leq U_g(X_t, k_g(t)), \tag{41}$$

where the uncertainty bonus is defined as:

$$U_g(X_t, k_g(t)) = \sqrt{\frac{\theta \log t}{k_g(t)}} + \phi(t) \cdot \max_{(x, \cdot) \in \text{NN}_g(X_t, k_g(t))} \rho(X_t, x). \tag{42}$$

Suppose $V(t)^{\complement}$ holds. Then there exists some $g \in \mathcal{G}$ such that:

$$\left| \widehat{f}_g(X_t, k_g(t)) - f_g(X_t) \right| > \sqrt{\frac{\theta \log t}{k_g(t)}} + \phi(t) \cdot \max_{(x,\cdot) \in \mathrm{NN}_g(X_t, k_g(t))} \rho(X_t, x). \tag{43}$$

For each $s \in [1, t-1]$, define:

$$\varepsilon_s = \mathbb{1}\{(X_s, Y_s^g) \in \mathrm{NN}_g(X_t, k_g(t))\}, \tag{44}$$
$$Z_s = \varepsilon_s \cdot (Y_s^g - f_g(X_s)). \tag{45}$$

Then the $k$-NN estimate can be decomposed as:

$$\widehat{f}_g(X_t, k_g(t)) = \frac{1}{k_g(t)} \sum_{s=1}^{t-1} \varepsilon_s Y_s^g = \frac{1}{k_g(t)} \sum_{s=1}^{t-1} \varepsilon_s f_g(X_s) + \frac{1}{k_g(t)} \sum_{s=1}^{t-1} Z_s. \tag{46}$$

By the Lipschitz assumption (Assumption A.2), for all $s \in \mathrm{NN}_g(X_t, k_g(t))$:

$$|f_g(X_s) - f_g(X_t)| \leq \lambda \cdot \rho(X_s, X_t) \leq \lambda \cdot r_{g,k_g(t)}(t), \tag{47}$$

where $r_{g,k_g(t)}(t)$ denotes the distance from $X_t$ to its $k_g(t)$-th nearest neighbor in $H_g(t)$. This implies:

$$\left| \frac{1}{k_g(t)} \sum_{s=1}^{t-1} \varepsilon_s f_g(X_s) - f_g(X_t) \right| \leq \lambda \cdot r_{g,k_g(t)}(t). \tag{48}$$

Therefore:

$$\left| \widehat{f}_g(X_t, k_g(t)) - f_g(X_t) \right| \leq \left| \frac{1}{k_g(t)} \sum_{s=1}^{t-1} Z_s \right| + \lambda \cdot r_{g,k_g(t)}(t). \tag{49}$$

If $V(t)^{\complement}$ holds, then:

$$\left| \sum_{s=1}^{t-1} Z_s \right| > \sqrt{\theta \log t \cdot k_g(t)}. \tag{50}$$

Since $Z_s$ are conditionally sub-Gaussian and zero-mean (Assumption A.4), we apply the inequality proved in Lemma A.9:

$$\mathbb{P}\left( \left| \sum_{s=1}^{t-1} Z_s \right| > \sqrt{\theta \log t \cdot k_g(t)} \right) \leq C \cdot t^{-\theta/2}, \tag{51}$$

for some constant $C$ depending on $\theta$. Taking a union bound over all $g \in \mathcal{G}$ and all $t \in [1, T]$ gives:

$$\sum_{t=1}^{T} \mathbb{P}(V(t)^{\complement}) \leq G \sum_{t=1}^{T} C \cdot t^{-\theta/2} < \infty, \tag{52}$$

which implies:

$$\sum_{t=1}^{T} \mathbb{E}[\mathbb{1}_{V(t)^{\complement}}] = O(1). \tag{53}$$

$\square$

## A.4 Concentration inequality

We next state and prove the Bernstein-type concentration inequality used in Lemma A.8.

**Lemma A.9.** *Fix a model $g$ and a round $t > G$. Recall that*

$$\varepsilon_s = \mathbb{1}\{g_s = g\} \, \mathbb{1}\{(X_s, Y_s^g) \in \mathrm{NN}_g(X_t, k_g(t))\}, \quad Z_s = Y_s^g - f_g(X_s),$$

*and $k = k_g(t) = \sum_{s=1}^{t-1} \varepsilon_s$. Under Assumption A.4 (sub-Gaussian noise), for any $\theta > 0$,*

$$\mathbb{P}\left( \left| \sum_{s=1}^{t-1} \varepsilon_s Z_s \right| > \sqrt{\theta \log t \, k} \right) \leq 2 \, t^{-\theta/2}.$$

*Proof.* Let $\{\mathcal{F}_s\}$ be the natural filtration generated by $(X_1, Y_1), \ldots, (X_s, Y_s)$. By construction $\varepsilon_s$ is $\mathcal{F}_{s-1}$–measurable and $Z_s$ is independent of $\mathcal{F}_{s-1}$. Moreover under Assumption A.4,

$$\mathbb{E}\big[e^{\rho Z_s} \mid \mathcal{F}_{s-1}\big] \leq \exp\big(\tfrac{\rho^2}{2}\big) \quad \forall \rho \in \mathbb{R}. \tag{54}$$

For any $\rho > 0$ define the process

$$W_s(\rho) = \exp\Big(\rho \sum_{u=1}^{s} \varepsilon_u Z_u - \tfrac{\rho^2}{2} \sum_{u=1}^{s} \varepsilon_u\Big), \quad W_0(\rho) = 1. \tag{55}$$

Then

$$\mathbb{E}\big[W_s(\rho) \mid \mathcal{F}_{s-1}\big] = W_{s-1}(\rho) \, \mathbb{E}\Big[e^{\rho \varepsilon_s Z_s - \frac{\rho^2}{2}\varepsilon_s} \,\Big|\, \mathcal{F}_{s-1}\Big].$$

Since $\varepsilon_s \in \{0, 1\}$ is $\mathcal{F}_{s-1}$–measurable,

$$\mathbb{E}\big[e^{\rho \varepsilon_s Z_s} \mid \mathcal{F}_{s-1}\big] = (1 - \varepsilon_s) + \varepsilon_s \, \mathbb{E}[e^{\rho Z_s} \mid \mathcal{F}_{s-1}] \leq (1 - \varepsilon_s) + \varepsilon_s \, e^{\rho^2/2} = e^{\frac{\rho^2}{2}\varepsilon_s}.$$

Hence $\mathbb{E}[W_s(\rho) \mid \mathcal{F}_{s-1}] \leq W_{s-1}(\rho)$, so $\{W_s(\rho)\}$ is a supermartingale. By Markov's inequality, for any $\eta > 0$,

$$\mathbb{P}\big(W_{t-1}(\rho) > e^{\eta}\big) \leq e^{-\eta} \, \mathbb{E}[W_{t-1}(\rho)] \leq e^{-\eta}. \tag{56}$$

Now on the event

$$\sum_{s=1}^{t-1} \varepsilon_s Z_s > \sqrt{\theta \log t \, k}, \tag{57}$$

choose $\rho = \sqrt{\theta \log t \, / \, k}$ and set

$$\eta = \rho \sqrt{\theta \log t \, k} - \tfrac{\rho^2}{2} k = \tfrac{\theta}{2} \log t.$$

Then

$$W_{t-1}(\rho) = \exp\Big(\rho \sum_{s=1}^{t-1} \varepsilon_s Z_s - \tfrac{\rho^2}{2} k\Big) > \exp\Big(\rho \sqrt{\theta \log t \, k} - \tfrac{\rho^2}{2} k\Big) = e^{\frac{\theta}{2}\log t} = t^{\theta/2}. \tag{58}$$

Therefore

$$\mathbb{P}\Big(\sum_{s=1}^{t-1} \varepsilon_s Z_s > \sqrt{\theta \log t \, k}\Big) \leq \mathbb{P}\big(W_{t-1}(\rho) > t^{\theta/2}\big) \leq t^{-\theta/2}. \tag{59}$$

The same argument applies to the negative tail $\sum_{s=1}^{t-1} \varepsilon_s Z_s < -\sqrt{\theta \log t \, k}$. A union bound yields the stated result. $\qquad\square$

# B    Hyperparameter tuning

## B.1    Optimal value of $\theta$

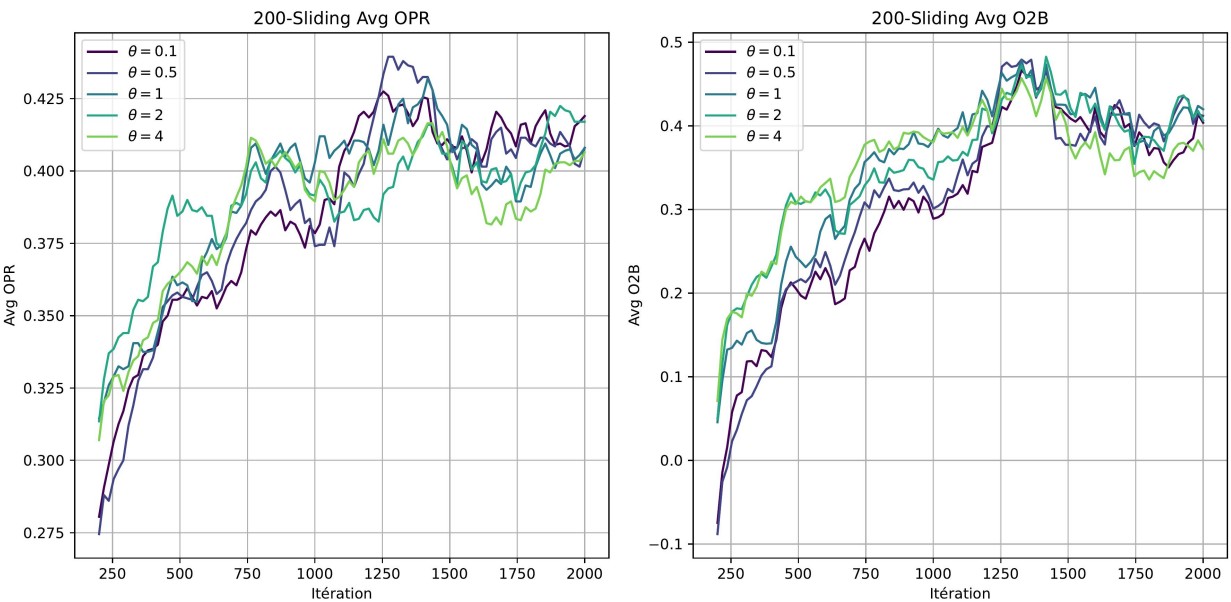

Figure 9: Performance of different $\theta$ values on the carrot-bowl dataset with 6 models. OPR (on the left), and OtB (on the right) are reported. Results are averaged over 10 runs.

Figure 9 reports the performance of our algorithm for different values of the UCB parameter $\theta$. Overall, the results show that the algorithm is relatively robust to the choice of this hyperparameter: performance varies only slightly across a wide range of values. In particular, $\theta$ values between 0.5 and 1 consistently yield strong performance. Based on this observation, we set $\theta = 1$ for all experiments.

## B.2    Thresholds calibration

This section details the calibration procedures used to set the threshold values for each variant described in subsection C.2 (except for the Warm-start variant which has no threshold).

In order to control the full-feedback budget in our active variants, we must set threshold parameters that determine when to trigger a query. Each variant relies on a different scoring mechanism—such as the top-two model gap (**Variant 1**), the local reward variance in the neighborhood (**Variant 4**), or the maximum UCB bonus (**Variant 2**)—and queries are triggered when this quantity falls below or exceeds a threshold.

To calibrate these thresholds meaningfully across datasets and budgets, we adopt a quantile-based strategy. Specifically, for each variant, we empirically compute the distribution of the associated quantity over a large set of prompts (e.g., 2,000 prompts from the Flickr dataset). Then, we determine the threshold as the $\alpha$-quantile of this distribution, where $\alpha$ reflects the target budget usage. For instance, setting $\alpha = 0.25$ will result in queries being triggered on roughly 25% of the prompts.

**Quantile curves.**    Figure 10 displays the quantile curves for all three variants, showing the value of the threshold as a function of $\alpha \in [0, 1]$. These curves are computed using the full validation set and reflect the empirical behavior of the scoring quantities.

**Practical usage.**    Given a desired budget ratio $\rho \in (0, 1)$ (e.g., $\rho = 0.2$ for 20% full feedback), we set the threshold for each variant to the $\rho$-quantile of the corresponding score distribution. This ensures that,

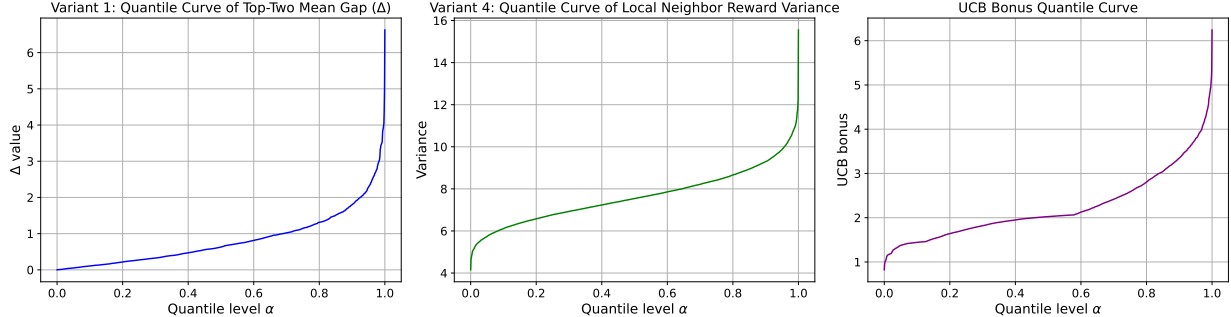

Figure 10: Quantile curves for each variant, plotting the threshold value as a function of the quantile level $\alpha$. These curves guide the choice of thresholds corresponding to a desired budget ratio.

on average, full-feedback queries are issued in only $\rho \cdot T$ rounds, where $T$ is the total number of iterations. The quantile curves provide a principled and interpretable method for aligning the budget with the scoring criteria used by each variant.

## C  Additional experimental details and results

### C.1  Inference Parameters for T2I Models

| Model | Resolution | Inference steps |
|---|---|---|
| Sana 1.5 | 1024×1024 | 18 |
| LCM Dreamshaper v7 | 768×768 | 50 |
| Unidiffuser v1 | 512x512 | 20 |
| SDXL-Turbo | 512×512 | 4 |
| SSD-1B | 1024×1024 | 50 |
| Koala-Lightning-700M | 1024×1024 | 25 |

Table 4: Recommended inference settings (resolution and number of steps) from each model's Hugging Face card and Diffusers defaults.

All models were run on a Nvidia RTX 3090 using Python 3.11 with Pytorch 2.7 for CUDA 12.8 on Ubuntu 22.04. All the model parameters used, including floating point precision, were the default ones from the huggingface library. CLIP Score was computed using the CLIP-ViT-L/14 model from the original paper (Radford et al., 2021).

### C.2  Active variant details

Formally, we implement each variant via a Boolean function $Q : X \in \mathcal{X} \rightarrow \{\text{True}, \text{False}\}$, chosen from one of four variants. In practice, we evaluate all four and select the best-performing strategy on a held-out prompt set (see subsection 5.2). Our regret analysis (Theorem 4.2) applies to the *Delta* variant.

**Variant 1: Delta (top-two gap).** This strategy triggers a full-query when the gap between the top two UCB indices is below a threshold $\delta$. Specifically, if $\hat{f}_{(1)}(X_t)$ and $\hat{f}_{(2)}(X_t)$ denote respectively the largest and second-largest estimates:

$$Q(X_t) = \text{True} \iff \hat{\Delta}(X_t) := \hat{f}_{(1)}(X_t) - \hat{f}_{(2)}(X_t) < \delta. \tag{60}$$

This criterion ensures that queries are concentrated in regions where the algorithm is "on the fence," i.e. where passive learning would struggle to confidently discriminate between competing models. It plays a

critical role in improving the convergence rate by providing decisive information at the points of highest ambiguity.

Crucially, this variant also allows us to improve the regret bound compared to passive algorithms. The Delta strategy ensures that the algorithm avoids spending too much time selecting suboptimal models in regions where the best model is clearly better. More precisely, we prove in Lemma A.6 any region of the prompt space where the gap $\Delta$ between the best model and the others is sufficiently large, the number of times a suboptimal model is chosen remains tightly controlled, scaling logarithmically in the time horizon $T$. Even though we do not have access to the true value of $\Delta$, we can compute its empirical estimate $\hat{\Delta}$. We show in Lemma A.7 that, in the long run, this estimate closely approximates the true gap, supporting its use.

**Variant 2: UCB-threshold.** Here, we query whenever the maximum *uncertainty bonus* across all models (the term $U_g(X_t)$ in the UCB index) exceeds a threshold $\varepsilon$. This captures situations of overall high variance in reward estimates:

$$Q(X_t) = \text{True} \iff \max_{g \in \mathcal{G}} U_g(X_t) > \varepsilon. \tag{61}$$

UCB-threshold tends to allocate queries to regions of the prompt space that are sparsely sampled, enforcing exploration of under-represented contexts.

**Variant 3: Warm-start.** A simple baseline: devote the first $B(T)$ rounds to full-feedback queries, then revert to passive KNN-UCB. This "bootstrap" strategy can be effective, as it allows the algorithm to leverage the information gained from early queries throughout the entire run:

$$Q(X_t) = \text{True} \iff t \leq B(T). \tag{62}$$

Warm-start front-loads the budget to rapidly seed each model's neighbourhood with diverse observations.

**Variant 4: Variance-threshold.** Finally, we query when the empirical variance of the $k_{g_t}(t)$ neighbours' rewards for the selected arm exceeds a threshold $v$. High local variance indicates that similar prompts have produced inconsistent rewards, suggesting that further full-feedback would clarify the true reward surface:

$$Q(X_t) = \text{True} \iff \text{Var}\left(\left\{y \mid (x, y) \in \text{NN}_{g_t}(X_t, k_{g_t}(t))\right\}\right) > v. \tag{63}$$

The procedures for selecting the thresholds $\delta$, $\varepsilon$, and $v$ are provided in subsection B.2.

#### Observation and updates

- If $Q(X_t) = \text{True}$ and $B > 0$, we observe the full reward vector $\{Y_t^g\}_{g \in \mathcal{G}}$, update each $H_g(t) \leftarrow H_g(t) \cup \{(X_t, Y_t^g)\}$, increment $N_g(t)$, and decrement $B \leftarrow B - 1$ (lines 12–14).

- Otherwise, we observe only $Y_t^{g_t}$ (lines 15–16) and update $H_{g_t}(t) \leftarrow H_{g_t}(t) \cup \{(X_t, Y_t^{g_t})\}$ and $N_{g_t}(t)$ accordingly.

By comparing these four strategies empirically, we identify which uncertainty signal best balances exploration and budget usage in diverse prompt distributions (subsection 5.2).

### C.3 CLIPScore

In all our experiments, we use a single metric to evaluate the quality of generated images: the CLIP-Score (Hessel et al., 2021). This metric is based on the CLIP embedding framework (Radford et al., 2021), which maps both the input prompt and the generated image into a shared embedding space. This enables a direct measurement of alignment between the textual and visual representations.

Formally, the CLIPScore between a prompt and a generated image is defined as:

$$\text{CLIPScore}(X, Y) = \max\left(0, \, 100 \cdot \cos(X, Y)\right), \tag{64}$$

where $X$ and $Y$ are the CLIP embeddings of the prompt and the generated image, respectively.

The CLIPScore thus reflects how well the semantic content of the generated image matches the input prompt, with higher values indicating stronger alignment.

*Remark* C.1. As we will observe in the experiments, CLIPScore is far from being a perfect evaluation metric. While it performs reasonably well at distinguishing poor-quality generations from clearly relevant ones, it often fails to discriminate between high-quality images produced by different state-of-the-art models. In practice, state-of-the-art models tend to achieve very similar CLIPScores, making them particularly hard to distinguish based on this metric alone. In particular, images that are judged by humans as less aligned with the prompt may sometimes receive a higher CLIPScore than better-aligned alternatives. However, this limitation is not critical for our study, as our algorithm is agnostic to the choice of evaluation metric and can operate with any scalar reward function.

### C.4 Effect of Sampling on CLIPScore Estimation

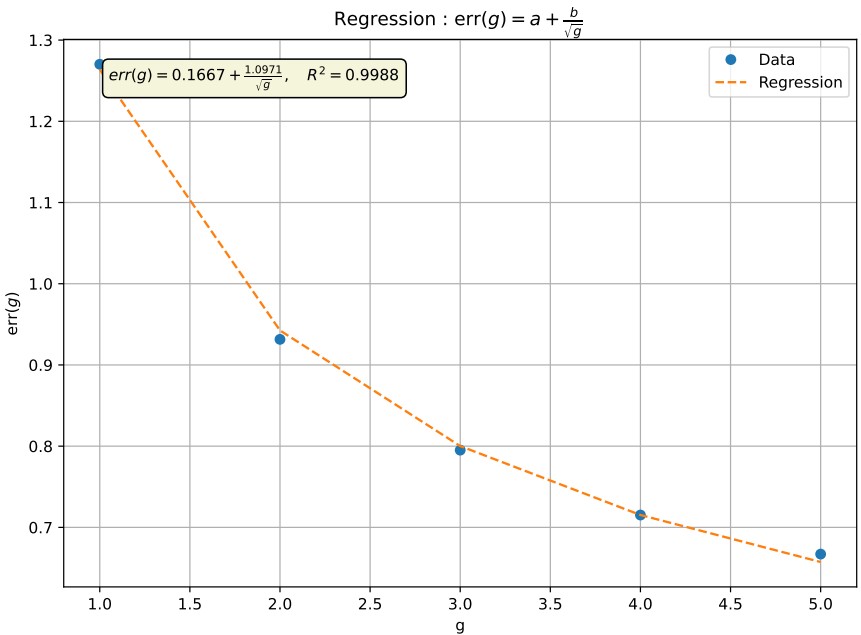

Figure 11: Regression plot of the error with respect to $g$

In this section, we study the convergence behavior of our algorithm with respect to the hyperparameter $g$, which denotes the number of samples used to compute the CLIPScore. In our experimental setup, the reward for each prompt-model pair is defined as the average CLIPScore over $g = 5$ independently generated images. Averaging over multiple generations reduces the variance of the reward signal and improves stability during training. However, it also implies that even an oracle algorithm, which always selects the model with the highest expected CLIPScore per prompt, cannot deterministically achieve the true optimum. This is because the observed reward is a finite-sample estimate of the model's mean performance, and thus inherently noisy.

We can therefore decompose the expected error of our algorithm, defined as the difference between its achieved CLIPScore and the true per-prompt optimum, as a function of $g$:

$$\text{err}(g) = a + \frac{b}{\sqrt{g}}, \tag{65}$$

for some constants $a, b \geq 0$. The $g^{-1/2}$ decay reflects the standard Monte Carlo convergence rate for the estimation of a mean from $g$ i.i.d. samples. Here, $a$ captures the irreducible approximation error of the

algorithm in the zero-variance limit (i.e., as $g \to \infty$), while $b$ quantifies the effect of noise due to finite sampling.

A regression analysis presented in Figure 11, for $g \in \{1, 2, 3, 4, 5\}$, illustrates this trade-off and allows us to estimate the values of $a$ and $b$. The points in the plot correspond to the final OtB values achieved by our algorithm on the MS-COCO dataset after $T = 5000$ iterations, for each value of $g$. Fitting the model $\mathrm{err}(g) = a + \frac{b}{\sqrt{g}}$ to the data yields an estimate of $a = 0.17$, indicating that our algorithm remains on average only 0.17 CLIPScore points below the oracle. This gap reflects the intrinsic approximation limit of our algorithm, independent of sampling noise. We expect this constant to decrease as the number of iterations $T$ increases, since more training steps allow the algorithm to better explore and exploit the prompt space.

## C.5   Analysis of distance/CLIPScore correlation

To better understand the relationship between the semantic similarity of prompts and the variability in their associated CLIP scores, we compute the cosine distance between all pairs of prompts (using CLIP text embeddings) and measure the absolute difference in their mean CLIP scores. We then discretize the distance range $[0, 1]$ into small bins (of width 0.01) and calculate the average CLIP score difference for each bin.

Figure 12 shows the resulting curve for the `SDXL-Turbo` model. As expected, prompt pairs that are semantically close (low cosine distance) tend to exhibit lower differences in CLIP scores, whereas more distant prompts show increasingly larger score variations. However, the correlation is not strictly linear: beyond a certain distance (around 0.4-0.6), the average CLIP score difference plateaus, indicating that highly dissimilar prompts do not necessarily lead to arbitrarily high score discrepancies. This suggests that semantic similarity is a useful but imperfect predictor of CLIP score differences, with additional factors (e.g., prompt structure or model-specific biases) contributing to variability.

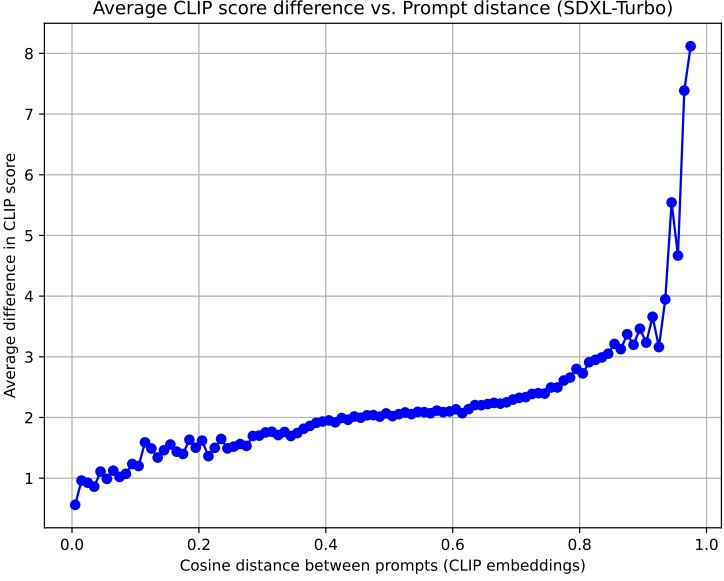

Figure 12: Average absolute difference in CLIP scores as a function of cosine distance between prompts, computed over 200k sampled prompt pairs for the `SDXL-Turbo` model on the `Flowers` dataset.

This analysis supports the intuition that closer prompts are more likely to have similar quality scores, justifying the use of nearest-neighbor methods for estimating expected reward in our bandit algorithms. Nonetheless, the observed noise and plateau region highlight the limitations of relying solely on prompt distance for score prediction.

### C.5.1 Close prompts mostly share the optimal model

Because the Lipschitz assumption (Assumption A.2) constrains the *expected* reward and not the realized CLIPScore, close prompts usually keep the same optimal model, and the rare flips are near-ties resolved by sampling noise rather than genuine smoothness violations. Figure 13 shows two close MS-COCO prompt pairs ($\rho \approx 0.10$): Pair 1 preserves the optimal model, while Pair 2 shows a noise-driven flip.

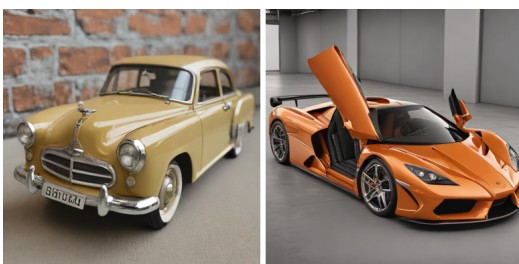

(a) **Pair 1**, $\rho = 0.098$. *Left:* "automobile model is in like new condition" (optimal model SDXL-Turbo, 28.12). *Right:* "automobile model is one of the most legendary supercars in the world" (optimal model SDXL-Turbo, 31.02). The optimal model is the same on both prompts.

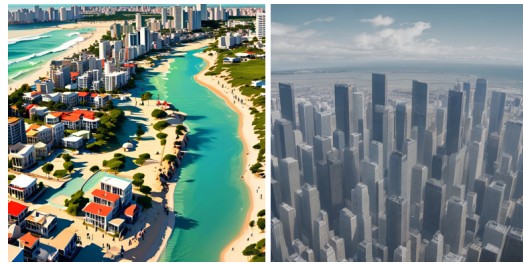

(b) **Pair 2**, $\rho = 0.100$. *Left:* "a city is a town on the eastern coast a resort with long stretches of sandy beach." (optimal model Koala, 31.20). *Right:* "a city on the west coast." (optimal model LCM, 28.15). The optimal model nominally flips Koala→LCM, but the leading models differ by less than one CLIPScore point (a near-tie resolved by noise).

Figure 13: Two close MS-COCO prompt pairs, each prompt rendered by its own optimal model. Near-identical prompts ($\rho \approx 0.10$) keep the same optimal model (Pair 1) or flip it only within the per-prompt sampling noise (Pair 2), consistent with the Lipschitz condition holding on the *expected* reward (Assumption A.2).

### C.6 Empirical verification of the Tsybakov margin condition

Our regret guarantee (Theorem 4.2) relies on the Tsybakov margin condition (Assumption A.3), which bounds the mass of prompts where two or more models are nearly tied: $\mu(\{0 < \Delta(x) < \delta\}) \leq C_\alpha \delta^\alpha$. We check this directly on the data. For each dataset we compute the per-prompt margin $\Delta(x) = \min\{\Delta_g(x) : \Delta_g(x) > 0\}$ from the measured mean rewards, form the empirical margin function $G(\delta) = \mu(\{0 < \Delta(x) < \delta\})$, and fit a power law $C_\alpha \delta^\alpha$. Figure 14 shows that $G(\delta)$ is closely matched by the fitted bound, with a strictly positive exponent $\alpha \approx 0.78$ on Carrot-bowl and $\alpha \approx 0.83$ on Flowers. The condition therefore holds empirically on real prompt distributions, and the fitted exponents are consistent with the values of $\alpha$ used in Table 10.

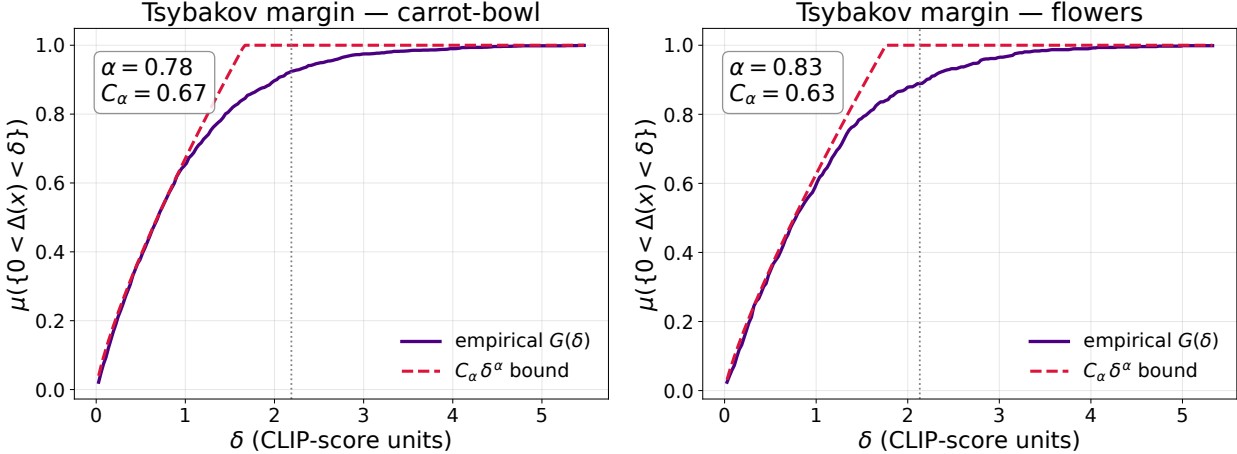

Figure 14: Empirical verification of the Tsybakov margin condition (Assumption A.3). For Carrot-bowl (left) and Flowers (right), the empirical near-tie mass $G(\delta) = \mu(\{0 < \Delta(x) < \delta\})$ (solid) is well described by the fitted bound $C_\alpha \delta^\alpha$ (dashed), with $\alpha \approx 0.78$ and $\alpha \approx 0.83$ respectively.

## C.7   Ablation studies figures

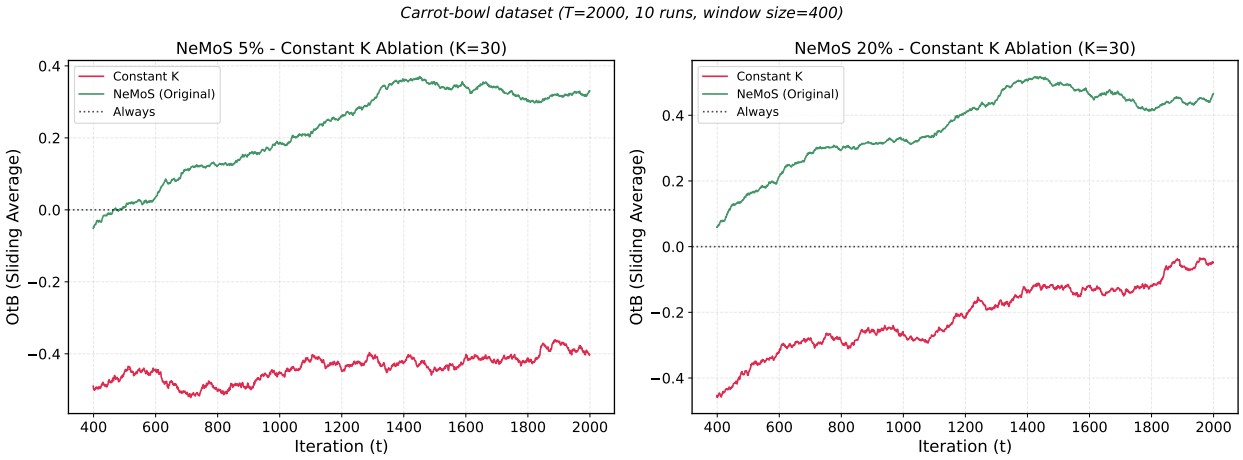

Figure 15: Constant K ablation study: Comparison between NeMoS with constant K=30 (this value achieved the best performance over the grid $\{10, 20, \ldots, 100\}$) versus original adaptive K selection. Results show sliding average OtB over 2000 iterations on Carrot-bowl dataset, averaged over 10 runs with window size 400.

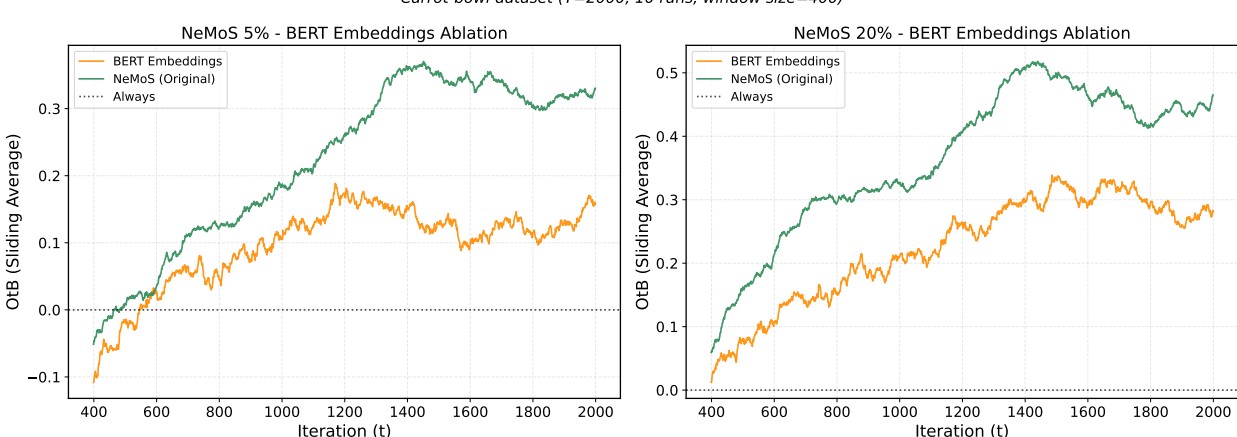

Figure 16: BERT embeddings ablation study: Comparison between **NeMoS** with BERT embeddings versus original CLIP embeddings. Results show sliding average OtB over 2000 iterations on Carrot-bowl dataset, averaged over 10 runs with window size 400.

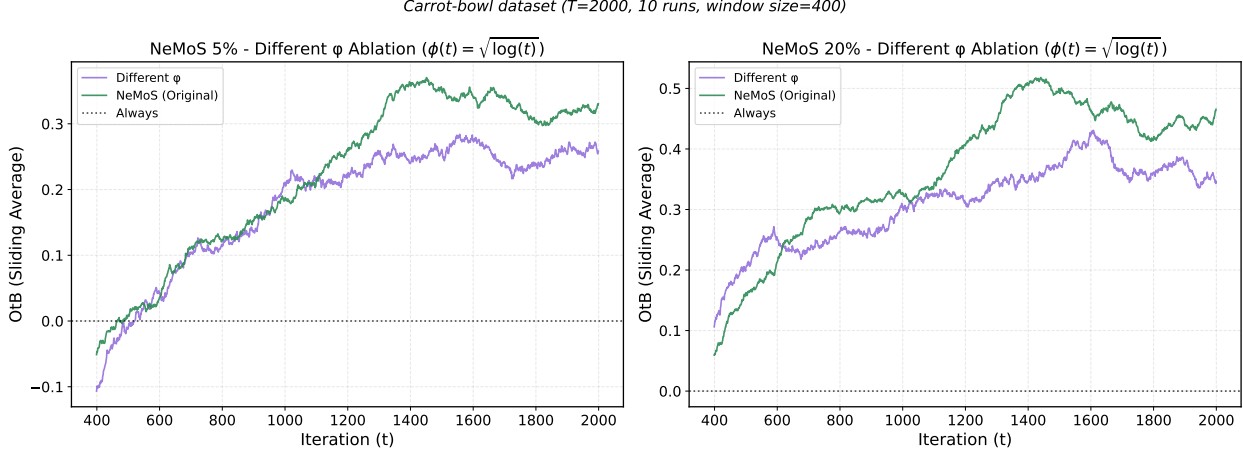

Figure 17: Different $\phi$ ablation study: Comparison between **NeMoS** with $\phi(t) = \sqrt{\log(t)}$ versus original exploration function ($\phi(t) = \log(t)$). Results show sliding average OtB over 2000 iterations on Carrot-bowl dataset, averaged over 10 runs with window size 400.

Table 5: GPU runtime comparison across baseline algorithms on Carrot-bowl dataset (T=2000). Runtime includes inference time for selected models and additional time when active queries are issued.

| Algorithm | Runtime (minutes) |
|---|---|
| Optimal | 18.33 |
| Always | 33.33 |
| Random | 17.74 |
| PAK-UCB | 17.73 |
| KNN-UCB | 20.72 |
| LinUCB | 18.78 |
| Neuronal-S | 38.96 |
| NeMoS 5% | 24.54 |
| NeMoS 20% | 40.06 |

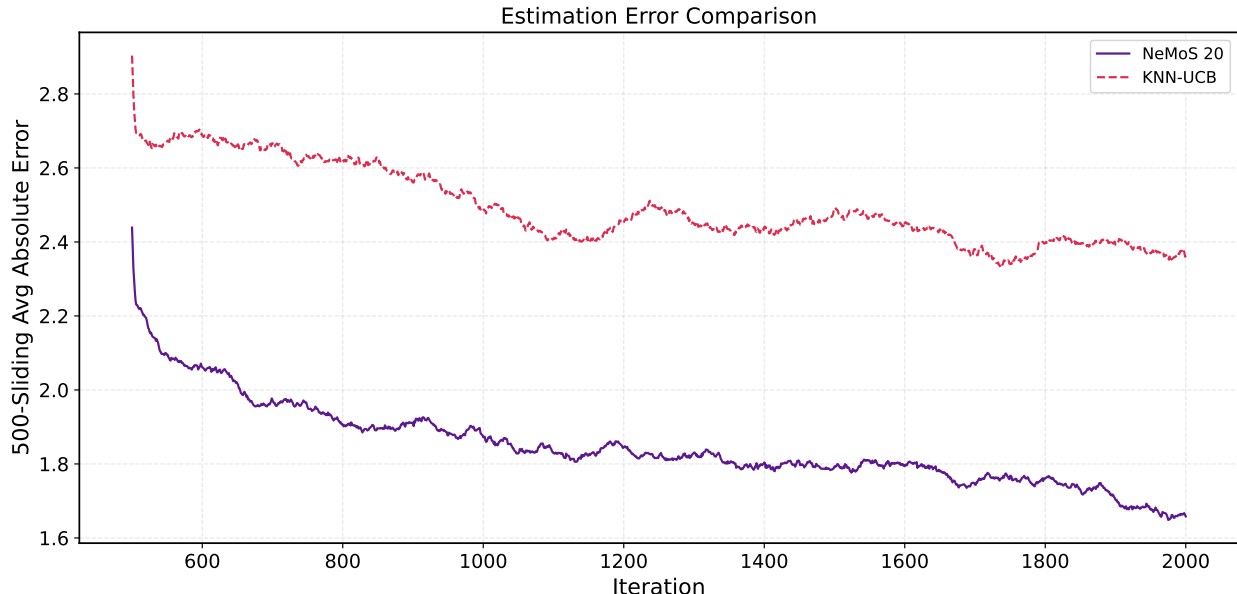

Figure 18: Intermediate results ablation study: Performance comparison of NeMoS and KNN-UCB showing average estimation error on Carrot-bowl dataset (T=2000, averaged over 10 runs). Active learning queries effectively accelerate the convergence of reward estimates.

The runtimes in Table 5 are dominated by model inference. To make the cost of the nearest-neighbor selection itself explicit, we separate it from inference. The per-step selection cost is $O(N_g(t))$ (the cumulative $O(G\,T^2 \log T)$ of subsubsection 4.3.2), but empirically it is a fixed overhead plus a tiny term linear in the stored history: the per-step wall-clock is well described by $\approx 5 \text{ ms} + 1.3 \times 10^{-4} \text{ ms} \times N_g(t)$, so the cumulative selection time follows a two-term $a\,T + b\,T^2$ shape rather than a single power law. At the horizons we run, the linear-overhead term dominates and the quadratic term is negligible: Table 6 reports that across three datasets and both budgets, kNN selection accounts for only 0.33–0.53% of NeMoS's total GPU runtime. The two contributions would only reach parity around $T \approx 4.3 \times 10^6$; at every realistic horizon, model inference dominates the wall-clock, so including the selection cost leaves Table 5 essentially unchanged.

Table 6: Share of NeMoS's GPU runtime spent on kNN selection versus model inference ($T = 2000$, 5 seeds, RTX 3090), on the three datasets that share the same six candidate models. **kNN compute** is the cuda-synced wall-clock of `select_arm`+`update`; **model inference** is the Table 5 attribution. The selection overhead is a sub-1% sliver on every dataset.

| Dataset | NeMoS | Model inference (min) | kNN compute (min) | kNN share |
|---|---|---|---|---|
| Carrot-bowl | 20% | 42.09 | 0.148 | **0.35%** |
| Carrot-bowl | 5% | 26.01 | 0.139 | **0.53%** |
| Flowers | 20% | 42.14 | 0.147 | **0.35%** |
| Flowers | 5% | 26.49 | 0.138 | **0.52%** |
| Flickr | 20% | 45.80 | 0.150 | **0.33%** |
| Flickr | 5% | 28.43 | 0.139 | **0.49%** |

Table 7: Delta ($\delta$) analysis showing average regret for different exploration parameter values on Carrot-bowl dataset (T=2000, averaged over 10 runs). Average Regret = Cumulative Regret / T.

| $\delta$ | Average Regret |
|---|---|
| $\delta = 0.20$ | $0.875 \pm 0.024$ |
| $\delta = 0.25$ | $0.850 \pm 0.018$ |
| $\delta = 0.30$ | $0.855 \pm 0.019$ |
| $\delta = 0.35$ | $\mathbf{0.804 \pm 0.018}$ |
| $\delta = 0.40$ | $0.839 \pm 0.015$ |
| $\delta = 0.45$ | $0.807 \pm 0.009$ |

Table 8: ImageReward experiment results showing average regret on Carrot-bowl dataset (T=2000, averaged over 10 runs). Comparison of all baseline algorithms using ImageReward-based evaluation metric.

| Algorithm | Average Regret |
|---|---|
| Optimal | $0.000 \pm 0.000$ |
| **NeMoS 20%** | $\mathbf{0.249 \pm 0.003}$ |
| NeMoS 5% | $0.319 \pm 0.007$ |
| KNN-UCB | $0.347 \pm 0.008$ |
| LinUCB | $0.400 \pm 0.012$ |
| Always | $0.432 \pm 0.005$ |
| Neuronal-S | $0.448 \pm 0.016$ |
| PAK-UCB | $0.474 \pm 0.014$ |
| Random | $0.489 \pm 0.019$ |

## C.8   Robustness to a third reward metric: HPSv2

To further confirm that our conclusions are not an artifact of CLIPScore (or of ImageReward), we re-run the baseline comparison on Carrot-bowl using **HPSv2** (Wu et al., 2023), a learned human-preference score, as the reward. We keep the two strongest contextual baselines (PAK-UCB and the passive KNN-UCB) and report results over 20 runs. As shown in Table 9 and Figure 19, NeMoS again attains the lowest cumulative regret (65.0 at a 20% budget and 69.8 at 5%), ahead of KNN-UCB (71.3) and PAK-UCB (72.9); the ranking is identical to the CLIPScore and ImageReward settings. For this experiment we use a time-scaled query threshold $\delta(t) \propto \log(t+1)$ so that the budget is consumed evenly over the horizon (visible in the budget panel of Figure 19, with cumulative consumption tracking the diagonal), rather than being front-loaded.

Table 9: HPSv2 experiment: final cumulative regret on Carrot-bowl ($T = 2000$, averaged over 20 runs). Lower is better. NeMoS achieves the lowest regret under this learned human-preference metric, matching the CLIPScore and ImageReward rankings.

| Algorithm | Cumulative regret |
|---|---|
| **NeMoS 20%** | **65.0** |
| NeMoS 5% | 69.8 |
| KNN-UCB | 71.3 |
| PAK-UCB | 72.9 |

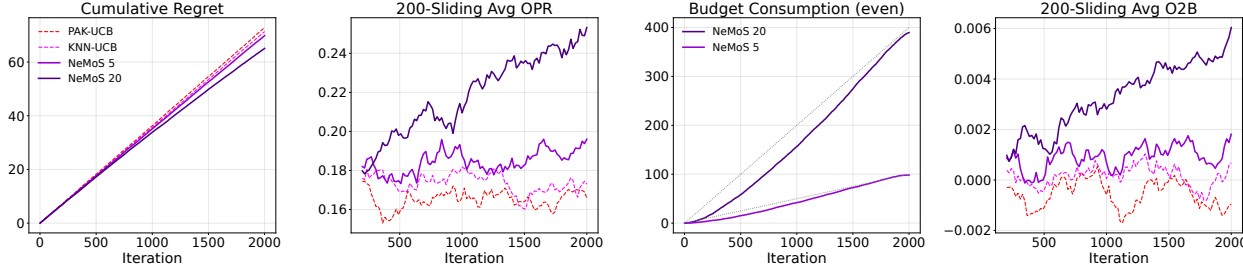

Figure 19: Baseline comparison on Carrot-bowl with the HPSv2 human-preference reward (20 runs). From left to right: cumulative regret, sliding-average OPR, budget consumption, and sliding-average OtB. NeMoS attains the lowest regret and highest OPR, and (budget panel) consumes its query budget evenly across the horizon thanks to the time-scaled threshold $\delta(t) \propto \log(t+1)$.

### C.9 Additional tables and figures

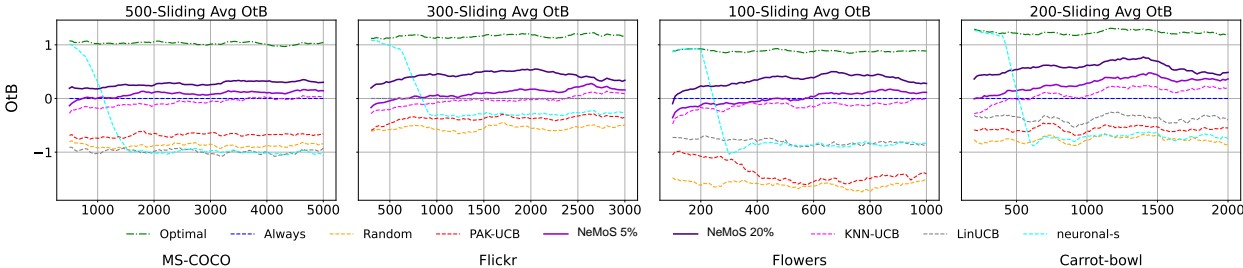

Figure 20: Sliding average OtB comparison between our algorithm and baselines across four prompt datasets with 6 models when using the query results to select. Results are averaged over 10 runs.

Table 10: Estimated values of $\alpha$, $d$, and $(d+2)/\alpha$. $d$ is estimated with the maximum-likelihood method of Levina & Bickel (2004) (averaged over $k = 10$–20 neighbours), and $\alpha$ via a logarithmic regression of the empirical margin function, consistent with the fits in subsection C.6.

| Dataset | $\alpha$ | $d$ | $(d+2)/\alpha$ |
|---|---|---|---|
| MS-COCO | 0.66 | 23.1 | 38.03 |
| Flickr | 0.73 | 21.1 | 31.64 |
| Carrot-Bowl | 0.78 | 16.8 | 24.10 |
| Flowers | 0.83 | 6.7 | 10.48 |

Table 11: Average total regret in the model addition setup.

| Algorithm | MS-COCO | Carrot-Bowl |
|---|---|---|
| Random | 1.542 | 1.668 |
| PAK-UCB | 1.378 | 1.387 |
| KNN-UCB | 0.868 | 0.880 |
| **NeMoS** | **0.734** | **0.709** |

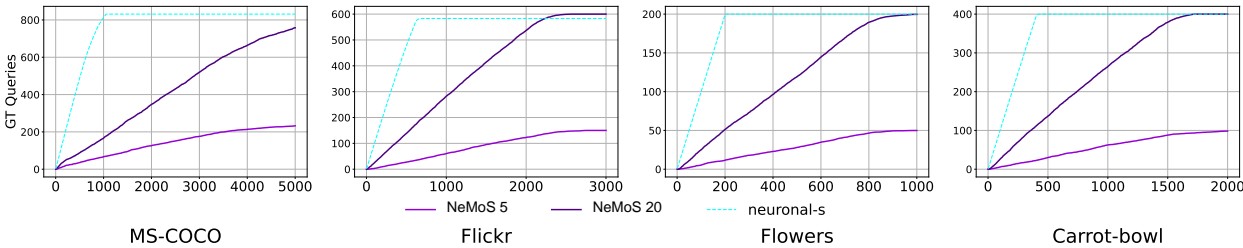

Figure 21: Budget consumption of the active algorithms shown in Figure 3 across the four datasets. NeMoS effectively distributes its budget over the entire horizon to maximize learning efficiency.

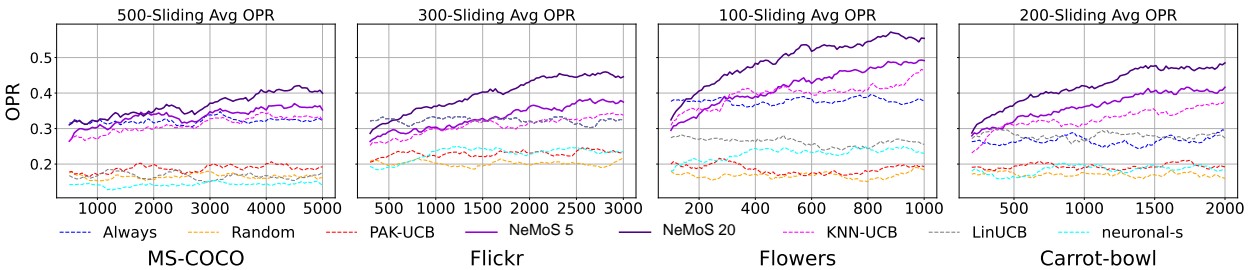

Figure 22: OPR plots corresponding to Figure 3.

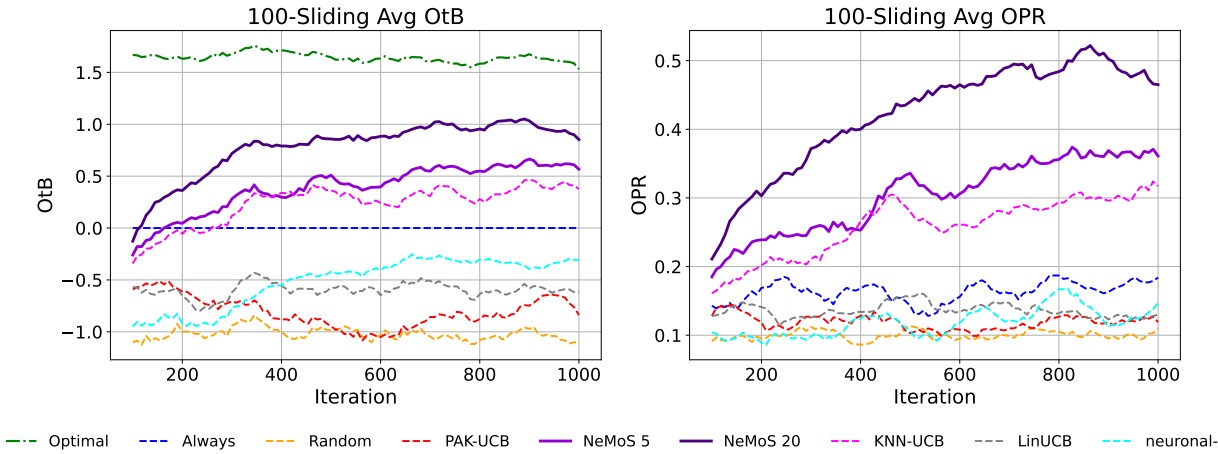

Figure 23: Sliding average OtB comparison between our algorithm and baselines across the Flowers dataset with **10** models. Results are averaged over 10 runs.

