# OpenReview forum: "NeMoS: Nearest Neighbors Bandit meets Active Learning for Online Model Selection"
_TMLR — Decision pending for TMLR_

### Review · Reviewer_kWXK · 2026-05-15

**Summary Of Contributions:**

This paper proposes NeMoS, an online prompt-wise model selection method for generative models. The method combines non-parametric nearest-neighbor reward estimation in prompt embedding space with a budgeted active querying rule that requests full feedback when the top candidate models are close. The paper provides a regret analysis and evaluates the method on four text-to-image prompt datasets.

Key strengths are the clean framing of prompt-wise generative model selection as an online bandit problem, the simple nearest-neighbor plus active-query mechanism, and the reasonably broad empirical experiments

The main weaknesses are that the scalability claim is not fully supported by the reported complexity/runtime evidence, the strongest empirical evidence still relies mostly on automatic scalar rewards such as CLIPScore, and the language-model/generalization claim is supported by a relatively small experiment.

**Audience:**

Yes

**Audience Explanation:**

The problem is relevant to TMLR readers interested in contextual bandits, active learning, and practical deployment of multiple foundation models.

**Broader Impact Concerns:**

I do not see major broader-impact concerns requiring rejection.

**Claims And Evidence:**

Yes

**Claims Explanation:**

The main claims are mostly supported. The paper gives a clear algorithm, a formal regret result under stated assumptions, and experiments across multiple text-to-image datasets and model pools showing consistent regret reductions. The active querying claim is also supported.

However, some claims should be tightened. The “scalable” framing is only partially justified because the exact nearest-neighbor procedure has quadratic-in-horizon selection cost, and the reported runtime table appears to emphasize GPU inference rather than full end-to-end overhead. The generalization beyond text-to-image is promising but based on one CommonsenseQA setup with two LLMs. The paper should either add evidence for these broader claims or phrase them more conservatively.

**Requested Changes:**

- Clarify the relationship between the theorem and the experiments. The regret bound assumes a budget schedule such as B(T)=T/logT and bounded rewards, while the experiments use fixed percentages and CLIPScore/ImageReward-style rewards. The authors should state exactly which experimental settings instantiate the theorem and which are heuristic.

- Provide end-to-end runtime and memory measurements, including nearest-neighbor search overhead, not only model inference/GPU runtime. If this is not feasible, the “scalable” claim should be weakened.

- Strengthen or narrow the generalization claim beyond text-to-image. The CommonsenseQA experiment with two LLMs is useful, but it is too small to support broad claims about general generative model selection.

---

### Review · Reviewer_Hb62 · 2026-05-29

**Summary Of Contributions:**

**Summary:**

This paper presents NeMoS, an algorithm for prompt-wise selection of image generation methods, framed as a conditional bandit problem. The approach combines nearest-neighbor reward estimation with a budget-constrained active learning strategy. Within a specified budget, the method outperforms competing baselines across several datasets. In addition, the authors provide theoretical contributions, including a regret lower bound and time and space complexity analysis.

**Strengths:**

- The method is flexible and appears to deliver meaningful improvements with limited additional cost compared to competitors.
- The mathematical results are non-trivial. In particular, the lower bound, conditioned on sufficiently small $\alpha$ and $d$, is a welcome contribution.
- The writing is clear and the mathematical content is rigorous.

**Weaknesses:**

- While the authors acknowledge that CLIP scores are a somewhat weak estimator of similarity, a significant part of the method relies on them. CLIP is known to exhibit biases and hallucination, which weakens the strength of the experimental evaluation.
- Related to the previous point, there is ambiguity around the notions of "similar" and "diverse" prompts. Although these concepts are central to the method, no concrete examples are provided to illustrate them.
- The method is designed for the specific budget $B(T) = T / \log T$, which can be quite restrictive. For example, on COCO, this amounts to roughly $5000 / \log(5000) \approx 587$ queries. How does the method perform under smaller budgets?
- I found the regret bound difficult to interpret. The bound depends on an exponent of $(d+2)/\alpha + (d+2)$, and according to Table 8, the orders of magnitude involved are substantial.
- The paper claims that the method addresses generalizability, yet it is tested on only two LLMs.

**Audience:**

Yes

**Audience Explanation:**

Online model selection is an important and timely topic, given that inference cost is a central concern in current practice. The work should also be of interest to researchers in the multi-armed bandit community.

**Claims And Evidence:**

Yes

**Claims Explanation:**

The scope of the method is clearly defined, and substantial evidence is provided to demonstrate the performance of NeMoS within the stated setup. Furthermore, the method is supported by theoretical insights.

**Requested Changes:**

**Major:**

- Clarify what prompt similarity implies in practical use cases: provide concrete examples of prompts identified as "similar" and discuss what this means in practice. In particular, showcase possible counterintuitive failures where prompts deemed similar by CLIP scores lead to unexpected outcomes, highlighting potential caveats of relying on CLIP for this purpose.
- Explain the practical implications of the actual values of $(d+2)/\alpha + (d+2)$ observed in the existing use cases, and discuss what these magnitudes mean for the tightness and usefulness of the bound.

**Minor:**

- Provide additional insight into how the method behaves under lower budgets.
- Test on a broader range of LLMs, or temper the generalizability claims accordingly.

---

### Review · Reviewer_pTyX · 2026-06-02

**Summary Of Contributions:**

The paper addresses the following problem: prompt-aware routing among generative models. It combines KNN-based reward estimation with budgeted active querying. The method uses prompt embeddings and local reward estimates instead of a fixed parametric reward model. Full-feedback queries on ambiguous prompts are a sensible design choice. The experiments are fairly broad: four prompt datasets, six image-generation models, and a CommonsenseQA LLM setting.

**Additional Comments:**

Please check comments above for more detailed feedback. Overall, I think the paper has promise but needs stronger evidence and a more careful discussion of assumptions and limitations.

**Audience:**

Yes

**Audience Explanation:**

* The problem is relevant to a growing set of ML systems that route prompts across multiple generative models.
* Researchers working on contextual bandits, model selection, prompt routing, generative-model evaluation, and adaptive inference would likely find the paper useful.
* Even if the current evidence needs strengthening, the framing and empirical setup should interest part of the TMLR audience.

**Broader Impact Concerns:**

I do not see major concerns.

**Claims And Evidence:**

No

**Claims Explanation:**

* The results are promising, but not fully convincing yet. The reported gains over baselines are fairly consistent across datasets and settings. Still, the evidence relies heavily on CLIPScore, even though the paper itself notes that this metric has important limitations. For a model-selection paper, this is a serious concern: if the reward signal is weak or noisy, the ranking conclusions may not reflect actual human preference.
* The theory is interesting, but it rests on strong assumptions that seem far from the practical setting. One concern is the Lipschitz reward assumption: the paper assumes that similar prompts should have similar expected rewards for a given model. This may not hold for generative models. Prompts that are close in embedding space can still differ in subtle semantics, style, or compositional structure, and those small changes can lead to very different outputs and reward values. I understand that this assumption follows prior work such as Reeve et al. (2018), but the paper should explain why it is reasonable for prompt-conditioned generative model selection. Right now, the theory and the practical application feel somewhat disconnected. The authors should either justify this assumption empirically, weaken it, or more clearly state that the regret guarantee applies only under an idealized smooth-reward setting. These assumptions may be useful for analysis, but the paper does not clearly show that they hold, even approximately, for real prompt distributions or modern generative models.
* Scalability is also under-supported. The nearest-neighbor procedure appears expensive, and the paper should better quantify runtime and memory costs for larger model pools and longer horizons. This concern is reinforced by the paper’s own complexity analysis, which gives an $O(GT^2 \log T)$ selection cost and $O(T + B(T) G)$ memory usage, suggesting that the method may become costly as the number of prompts or candidate models grows.
* Overall, the conclusions are directionally supported, but some claims about broad practical superiority feel too strong for the current evidence.

**Requested Changes:**

* Add stronger evidence beyond CLIPScore. Human preference evaluation, a few other automatic reward metric, or a careful metric-sensitivity study would make the empirical claims much more credible.
* Temper broad claims about practical superiority. The current results support promise, not yet general superiority across real deployment settings.
* Discuss the theoretical assumptions more honestly. The paper should explain when intrinsic-dimension, Lipschitz, and Tsybakov-style assumptions are plausible, and when they may fail.
* Provide a clearer scalability analysis. Report runtime, memory use, and behavior for larger model pools or longer horizons.
* Improve notation and definitions in the theory section. Some symbols and assumptions appear too quickly.
* Make the figures easier to read, especially dense appendix-style plots. Larger fonts, clearer axes, and shorter takeaway-focused captions would help.
* Add experiments where prompt embeddings are less aligned with reward, to test robustness.
* Study larger candidate model sets, since the current six-model setup may not reflect realistic routing scenarios.
* Add more discussion of failure cases, especially prompts where the active-query strategy does not help.

---

### Decision · Action_Editor_cU1W · 2026-07-03

**Recommendation:** Accept as is

**Additional Comments:**

I thank the reviewers and authors for the constructive review process. The paper presents a timely and relevant contribution on online prompt-wise model selection using nearest-neighbor contextual bandits with budgeted active querying. The initial reviews raised important concerns about the strength of the empirical evidence, the reliance on CLIPScore, the practical interpretation of the theoretical assumptions, scalability/runtime, and the breadth of the generalization claims. The revised manuscript substantially addresses these points through additional experiments, clearer discussion of assumptions and limitations, runtime measurements, and more careful framing.

The remaining concerns are now appropriately scoped rather than fatal. The paper should be understood as an appropriate TMLR contribution on online generative model selection, with promising but still limited evidence for broader generalization beyond the main text-to-image setting.

**Audience:**

Yes

**Audience Explanation:**

The submission is of interest to TMLR readers working on contextual bandits, active learning, adaptive inference, prompt routing, model selection, and evaluation/deployment of generative models. The problem of online prompt-wise model selection is timely, and the paper provides a clear algorithmic contribution together with both theoretical and empirical support. Even though the work does not clearly meet the higher bar for a major conference presentation track, it is relevant and useful for TMLR’s audience.

**Claims And Evidence:**

Yes

**Claims Explanation:**

The main claims are supported by the revised manuscript. The original reviews identified substantive concerns about reliance on CLIPScore, the relation between the theoretical assumptions and the empirical setting, scalability/runtime, and overly broad generalization claims. The authors addressed these concerns with additional experiments using HPSv2/ImageReward, empirical evidence for the Tsybakov-margin behavior, examples clarifying prompt-similarity limitations, runtime and scalability measurements, lower-budget experiments, a larger model-pool experiment, and more conservative language around generalization.

Some limitations remain: the theoretical guarantee depends on strong smoothness/margin assumptions, the regret bound is not necessarily tight or directly predictive at the experimental scales, and the evidence beyond text-to-image model selection is still limited. However, these limitations are now sufficiently discussed and scoped, and all three reviewers’ final recommendations are to accept. I therefore judge the central claims to be accurate and adequately supported.